# Diversity of *Alternaria* Section *Nimbya* in Iran, with the Description of Eight New Species

**DOI:** 10.3390/jof11030225

**Published:** 2025-03-15

**Authors:** Abdollah Ahmadpour, Youbert Ghosta, Zahra Alavi, Fatemeh Alavi, Alireza Poursafar, Pabulo Henrique Rampelotto

**Affiliations:** 1Higher Education Center of Shahid Bakeri, Urmia University, Miyandoab 59781-59111, Iran; 2Department of Plant Protection, Faculty of Agriculture, Urmia University, Urmia 57561-51818, Iran; 3Department of Plant Pathology, North Dakota State University, Fargo, ND 58102, USA; alireza.poursafar@ndsu.edu; 4Bioinformatics and Biostatistics Core Facility, Institute of Basic Health Sciences, Federal University of Rio Grande do Sul, Porto Alegre 91501-970, Brazil

**Keywords:** *Alternaria*, ISSR marker, multi-gene phylogeny, new taxa, species diversity, taxonomy

## Abstract

*Alternaria* includes endophytes, saprophytes, and pathogens affecting both plants and animals, with a global distribution across various hosts and substrates. It is categorized into 29 sections, each defined by a type species and six monophyletic lineages. The *Alternaria* section *Nimbya* comprises 10 species primarily associated with the families *Juncaceae* and *Cyperaceae*, functioning as either saprophytes or plant pathogens. In this study, 189 fungal strains were collected from multiple locations across six provinces in Iran. The isolates were initially classified based on morphological characteristics and ISSR-PCR molecular marker banding patterns. Multi-gene phylogenetic analyses of 38 selected strains, using ITS–rDNA, *GAPDH*, *TEF1, RPB2*, and *Alt a 1* gene sequences, combined with morphological data, led to the identification of 13 species, including eight new species, namely *Alternaria caricifolia*, *A. cyperi*, *A. juncigena*, *A. junci-inflexi*, *A. persica*, *A. schoenoplecti*, *A. salkadehensis*, and *A. urmiana*. In addition, this work identified new host associations (*matrix nova*) for three previously known species: *A. caricicola* on *Cyperus* sp., *A. cypericola* on *Eleocharis* sp., and *A. junci-acuti* on *Carex* sp. The study provides detailed morphological descriptions and illustrations of all identified species, discusses their habitats, distribution, and phylogenetic relationships within section *Nimbya*, and presents a key for species identification within this section in Iran. Furthermore, these findings highlight the significance of studying fungal biodiversity in Iran and contribute to a better understanding of species distribution and host range within the *Alternaria* section *Nimbya*.

## 1. Introduction

The genus *Nimbya* E.G. Simmons was introduced with *N. scirpicola* (Fuckel) E.G. Simmons as the type species to accommodate fungi with a morphology similar to the genera *Alternaria* Nees, *Sporidesmium* Link, and *Drechslera* S. Ito. However, the most distinct and useful diagnostic character for differentiating *Nimbya* from these genera is the internal compartmentation of conidia into well-defined, usually angular lumina, surrounded by an abundant distoseptum matrix [1]. The conidial lumina are interconnected by narrow channels through the transverse distosepta, with transverse eusepta inserted at maturity. Sexual morphs of *Nimbya* spp. were classified under the genus *Macrospora* Fuckel [1,2]. After the initial description of five *Nimbya* species, the number of described species, or those transferred to *Nimbya* from other genera based on morphological characteristics, gradually increased. By 2005, the number of *Nimbya* had reached 17 species [3,4,5,6,7,8,9]. Various species of *Nimbya* have been recorded on plants from the families *Amaranthaceae*, *Asteraceae*, *Caryophyllaceae*, *Cyperaceae*, *Euphorbiaceae*, *Fabaceae*, *Juncaceae*, and *Solanaceae* [1,9]. Based on conidium morphology, a close relationship between *Nimbya* and *Alternaria* was proposed [2,4].

With the advent of genome sequencing and phylogenetic analyses, these methods became valuable tools for identifying species boundaries and phylogenetic relationships. In the first phylogenetic study of *Nimbya*, the relationships among *N. caricis* and *N. scirpicola*, along with species of *Embellisa*, *Alternaria*, *Ulocladium*, and *Stemphylium*, were examined using the internal transcribed spacer region of rDNA (ITS–rDNA), the mitochondrial small-subunit (mt SSU), and glyceraldehydes-3-phosphate dehydrogenase (*GAPDH*) gene sequences [10]. The results revealed that the studied species in *Alternaria*, *Embellisia*, *Nimbya*, and *Ulocladium* formed a large monophyletic clade, with *Stemphylium* as a sister clade. Moreover, *Ulocladium* and *Embellisia* were found to be polyphyletic, suggesting that using only morphological characteristics to differentiate these genera could be misleading, as the key characters used are homoplastic [10]. Similar results from *GAPDH* and *Alternaria* major allergen (*Alt a 1*) genes sequences, also indicated close relationships between *Nimbya* spp., *Alternaria* spp. in the infectoria species group, and some *Embellisia* spp. [11]. A more comprehensive phylogenetic study using a larger set of isolates examined the relationships of *Nimbya* and *Embellisia* with other closely related genera, based on ITS–rDNA, *Alt a 1*, and *GAPDH* sequences [12]. The study revealed the polyphyletic nature of both *Embellisia* and *Nimbya*, dividing species within each genus into four and two clades, respectively. *Nimbya* group I included four species (*N. caricis*, *N. scirpicola*, *N. scirpinfestans*, and *N. scirpivora*), while group II contained three species (*N. alternantherae*, *N. celosiae*, and *N. perpunctulata*). Morphological and ecological comparisons among these species indicated that group I species had short-beaked conidia and were associated with the *Juncaceae* and *Cyperaceae* plant families, while the group II species, with long-beaked conidia, originated from plants in the family *Amarantaceae*, suggesting independent evolutionary origins. Three species from group II were transferred to *Alternaria*, and a new species group, *Alternanthera* species group, was established to accommodate them [12]. This species group concept was an early attempt at a classification scheme for identifying *Alternaria* species based on two morphological characteristics: three-dimensional sporulation pattern and conidium morphology, with each species group typified by a representative species [4,13]. In a subsequent study using five phylogenetically informative loci—*GAPDH*, *Alt a 1*, Actin (*ACT*), Plasma membrane ATPase (*ATPase*), and Calmodulin (*CAL*)—eight well-supported asexual lineages of *Alternaria* were identified and grouped into species groups (both previously known and newly identified) [14]. These eight species groups were elevated to the taxonomic rank of “section”, a rank that had previously been used by Neergaard [15] and Joly [16] to classify *Alternaria* species.

In a landmark study aimed at delineating the phylogenetic lineages and establishing a robust taxonomy within *Alternaria* and allied genera, six genomic regions (18S nrDNA (SSU), 28SnrDNA (LSU), ITS–rDNA, *GAPDH*, *RPB2*, and *TEF1*) were sequenced [17]. This study recognized 24 internal clades and six monotypic lineages under the *Alternaria* clade. The 24 clades were called sections (each typified by a type species), thirteen generic names, *Allewia*, *Brachycladium*, *Chalastospora*, *Chmelia*, *Crivellia*, *Embellisia*, *Lewia*, *Nimbya*, *Sinomyces*, *Tretispora*, *Ulocladium*, *Undifilum*, and *Ybotromyces* were put as synonymy with *Alternaria* and generic circumscription of *Alternaria* was amended. In addition, species previously classified as *Nimbya* proper (*Nimbya* group I) [12], were reclassified under *Alternaria* and placed in section *Nimbya*. A re-examination of morphological characteristics of species formerly identified as *Nimbya* spp., without phylogenetic analysis, to determine their affiliation with any *Alternaria* section, resulted in the inclusion of *A. heteroschemos* and *A. juncicola* in the *Nimbya* section [18]. More recently, four new species—*A. caricicola*, *A. cypericola*, *A. heyranica*, and *A. junci-acuti*—were described in the *Nimbya* section, based on combined morphological features and multi-gene phylogeny, and the number of species in this section was raised to 10 [19,20].

The plant order *Poales* is one of the most diverse and ecologically significant groups of angiosperms, encompassing 14 families and over 24,300 species. Members of the *Poales* families are found worldwide and are notable for their species richness, ecological adaptations, and diverse life forms. They have significantly influenced the evolution of mammals, including hominids, and form the foundation of much of the human diet [21,22]. Among these, the *Cyperaceae* family, commonly known as sedges, ranks as one of the three largest monocot families and one of the 10 largest plant families overall, with 5687 distributed globally [23]. *Cyperaceae* plants play crucial roles in various habitats, from wetlands to deserts, dominating wetland vegetation, contributing to nutrient cycling, and providing habitats for numerous species. Additionally, members of this family are used for medicinal purposes, to treat multiple diseases and ailments, as important sources of essential oils, and as a source of food and animal feed [24,25]. However, some *Cyperaceae* species are invasive weeds, posing threats to the natural ecosystems and agricultural and forest productivity [26]. The family *Juncaceae*, also known as the rush family, comprises around 500 species of monocotyledonous flowering plants. These plants are integral to aquatic and wetland ecosystems, offering habitats for fauna, helping slow water flow, preventing soil erosion, and improving water quality. Historically, certain *Juncaceae* species have been used for crafting materials such as baskets and chair bottoms, as well as in traditional herbal medicines due to their bioactive compounds, including phenanthrenes and phenanthrenes-like molecules [27,28].

Both *Cyperaceae* and *Juncaceae* plants serve as hosts for a wide variety of fungi, including members of the *Alternaria* section *Nimbya*. Some of these fungi are parasitic, causing substantial damage to their host plants, while others function as saprophytes, aiding in decomposition and nutrient cycling. Previous studies on *Alternaria* species associated with these plant families have already shown a significant level of species diversity [19,20]. In this study, a larger set of fungal isolates were collected from various regions across six provinces in Iran. Through a combination of morphological examination and multi-gene phylogenetic analysis, thirteen species were identified, eight of which are new to science. This paper provides detailed descriptions and illustrations of these species, as well as an analysis of their phylogenetic and host relationships within the section.

## 2. Materials and Methods

### 2.1. Fungal Isolates

Plant samples, including leaves and culms from the families *Cyperaceae* and *Juncaceae*, exhibiting discolored lesions and blight symptoms, were collected from various wetland areas across six provinces in Iran (Ardebil, East Azarbaijan, Golestan, Guilan, Mazandaran, and West Azarbaijan) during 2019–2021 (Figure 1). The samples were properly labeled, kept cold, and transported to the laboratory. Fungal isolation and purification were followed using the method described by Ahmadpour et al. [20]. Fungal isolates were kept in potato carrot agar (PCA, 20 g potato, 20 g carrot, 20 g agar, and 1000 mL of distilled water) slants at 4 °C for short-term preservation and on sterile filter paper segments at −20 °C for long-term preservation. Pure cultures of all identified isolates were deposited in the fungal culture collections of the Iranian Research Institute of Plant Protection (IRAN) and Urmia University (FCCUU).

### 2.2. Morphological Observations

Morphological observations were made from the fungal strains growing on potato carrot agar (PCA) medium, incubated at 23–25 °C under Cool White fluorescent light with 8/16 h light/dark cycle for 5–7 days without humidity control [1,2]. Strains showing reduced sporulation under these conditions were transferred to synthetic nutrient-poor agar medium (SNA) [29] containing a small piece of autoclaved filter paper to promote sporulation [17]. Slide mounts were prepared using lactophenol or lactophenol cotton blue, and micro-morphological features of hyphae, as well as asexual and sexual structures (when induced) were examined using an Olympus AX70 compound microscope (Olympus Optical CO., LTD, Tokyo, Japan). Measurements were taken for 30–50 structures, including hyphae, conidiophores, conidia, ascomata, asci, and ascospores, and microphotographs were captured from the slide mounts. Colony morphology was also characterized from cultures grown on Potato Dextrose Agar (PDA, Merck, Darmstadt, Germany), PCA, and V-8A (175 mL of commercial V8 vegetable juice, 3 g CaCO_3_, 20 g Agar, and 1000 mL of distilled water) media after incubation in the dark at 25 °C for 7 days. Colony color was determined using Rayner’s color charts [30]. To induce ascomata formation, 2% water agar plates containing a 6 cm autoclaved host plant culm segment were inoculated and incubated at three temperatures (15, 18, and 23 °C) for 30–60 days [8]. Taxonomic novelties were deposited in MycoBank [31].

### 2.3. DNA Extraction and PCR Amplification

DNA was extracted from the mycelial mass of each isolate, which was obtained from 10-day-old PDA plates, using a standard sodium dodecyl sulfate (SDS) lysis buffer. The extraction process involved chloroform extraction followed by isopropanol precipitation [19]. Polymorphic banding patterns were generated for all isolates using inter simple sequence repeat (ISSR)-PCR with the ISSR5 ((GA)5YC) primer for preliminary screening. Each PCR reaction contained 0.4 μM of the primer, 4 μL ready master mix (Taq DNA polymerase 2X Master Mix Red, 2 mM MgCl_2_, Ampliqon Company, Odense, Denmark), and approximately 10 ng of template DNA in a total volume of 10 μL. The thermal cycling conditions included an initial denaturation step at 95 °C for 5 min, followed by 35 cycles of 95 °C for 45 s, 41 °C for 60 s, and 72 °C for 90 s, with a final extension step at 72 °C for 10 min. Amplicons were visualized on a 1% agarose gel. Isolates with identical banding patterns were classified as closely related taxa [20]. To analyze phylogenetic relationships, 38 isolates were chosen based on ISSR banding pattern and morphological characteristics for multi-gene sequencing. Target regions included the internal transcribed spacer (ITS–rDNA), parts of the glyceraldehyde-3-phosphate dehydrogenase (*GAPDH*), RNA polymerase II largest subunit (*RPB2*), translation elongation factor 1-alpha (*TEF1*) and *Alternaria* major allergen (*Alt a 1*) genes, using primers listed in Table 1. PCR conditions and reaction mixtures were consistent with Ahmadpour et al. [20]. The resulting amplicons were visualized on a 1.5% agarose gel stained with GelRedTM (Biotium, Hayward, CA, USA) and viewed under UV light. Amplicon sizes were determined using a HyperLadderTM I molecular marker (Bioline, Memphis, TN, USA). The amplified products were then purified and sequenced using the same primer sets by Macrogen Corp. (Seoul, Republic of Korea).

### 2.4. Sequence Alignment and Phylogenetic Analyses

DNA sequences generated for the new strains were examined and trimmed using MEGA 6.0 [38], and then exported as FASTA files for subsequent analyses. Published and verified DNA sequences of various loci for the type or representative *Alternaria* strains were obtained from the GenBank database and incorporated into the phylogenetic analyses (Table 2) [12,17,19,20,39,40,41,42,43]. Multiple sequence alignments for each locus were created using the MAFFT version 7 online tool (https://mafft.cbrc.jp/alignment/server/; accessed on 20 February 2025) [44], with further manual adjustments and trimming conducted in MEGA 6.0 when needed. A five-gene concatenated dataset (ITS–rDNA + *GAPDH* + *TEF1* + *RPB2* + *Alt a 1*) was generated using Mesquite v. 3.61 [45]. Two separate multi-locus phylogenetic analyses were performed: one focusing on *Alternaria* species within section *Nimbya* using a concatenated five-gene dataset, and a broader analysis including DNA sequences from 29 *Alternaria* sections and six monotypic lineages. Bayesian inference (BI) analysis was conducted using MrBayes v. 3.2.7 [46] with the Markov Chain Monte Carlo (MCMC) method, utilizing four chains, 1M generations, and a heated chain temperature of 0.1. Trees were saved every 1000 generations, with a burn-in of 25%, and posterior probabilities (PP) calculated from the remaining trees. The run was considered complete when the average standard deviation of split frequencies dropped below 0.01. MrModeltest 2.3 [47] and the Akaike Information Criterion (AIC) were used to identify the best-fit evolutionary models for BI (Table 3). Maximum-likelihood analyses were performed with the RAxML-HPC BlackBox v. 8.2.12 [48] on the CIPRES Science Gateway version 3.3 (accessible at https://www.phylo.org/) [49], using the GTRGAMMA + I substitution model. Maximum Parsimony (MP) analysis was carried out in PAUP v. 4.0b10 [50], using a heuristic search with 1000 random sequence additions and tree-bisection-reconnection (TBR) branch swapping, treating gaps as missing data. Bootstrap support values were calculated from 1000 replicates, and descriptive tree statistics [Tree Length (TL), Consistency Index (CI), Retention Index (RI), and Homoplasy Index (HI)] were determined from MP analysis. Outgroup taxa included sequences of *Alternaria chlamydospora* (CBS 491.72), *A. phragmospora* (CBS 274.70) from section *Phragmosporae*, and *Stemphylium botryosum* (CBS 714.68) and *S. vesicarium* (CBS 191.86) for small- and large-scale analyses, respectively. The resulting phylogenetic trees were visualized in FigTree v. 1.4.4 [51] and refined using Adobe Illustrator^®^ CC 2021. The newly generated sequences were submitted to GenBank (Table 2) and the concatenated alignments were deposited in TreeBASE (https://www.treebase.org) under the Submission ID 26865.

### 2.5. Genealogical Concordance Phylogenetic Species Recognition Analysis

The Genealogical Concordance Phylogenetic Species Recognition (GCPSR) was employed to identify significant recombinant events [52]. Five-locus concatenated dataset (including ITS–rDNA, *GAPDH*, *TEF1*, *RPB2*, and *Alt a 1*) was used to determine the recombination level within phylogenetically closely related species. The data were analyzed using SplitsTree 5 software, applying the pairwise homoplasy index (PHI or Φw) test [53,54]. Results from the PHI test, with a value less than 0.05 (Φw < 0.05), indicate a significant presence of recombination within the dataset. To visualize the relationships between new taxa and their closely related species, split graphs were constructed and visualized using both the LogDet transformation and split decomposition options.

## 3. Results

### 3.1. Phylogenetic Analyses

A summary of phylogenetic information and substitution models for each dataset is provided in Table 3. The topologies of individual gene trees were consistent, with no conflicts observed in species delimitation. In the large-scale phylogenetic tree, combined analyses using BI and ML/MP methods revealed that our isolates grouped with strong support (ML/MP/BI = 99/74/1.0) within the *Alternaria* section *Nimbya*, distributed in 13 lineages (Figure 2). The multi-locus datasets for large-scale analyses included a total of 2299 characters (1257 constant sites, 1042 variable sites, 106 parsimony-uninformative sites, and 936 parsimony-informative sites), including gaps (462 for ITS, 524 for *GAPDH*, 187 for *TEF1*, 690 for *RPB2*, and 436 for *Alt a 1*) (Table 3). For the small-scale analyses, which focused exclusively on *Alternaria* species within section *Nimbya*, a total of 2307 characters from 55 strains were used (1610 constant sites, 697 variable sites, 52 parsimony-uninformative sites, and 645 parsimony-informative sites), including gaps (469 for ITS, 514 for *GAPDH*, 177 for *TEF1*, 716 for *RPB2*, and 431 for *Alt a 1*) (Table 3). In the *Alternaria* section *Nimbya* phylogenetic tree, our isolates formed 13 distinct lineages with high support values, indicating the presence of 13 species; five have been previously described and eight are newly identified (Figure 2 and Figure 3).

### 3.2. Taxonomy

A total of 189 fungal isolates were obtained from *Cyperaceae* and *Juncaceae* plants. All isolates were examined based on morphology, and ISSR-PCR molecular marker banding patterns, and 38 representative isolates were selected from different plant hosts for phylogenetic analyses (Appendix A). Based on phylogenetic analyses and morphological characteristics, the studied isolates were assigned to 13 species in the *Alternaria* section *Nimbya*, including eight new species (*A. caricifolia*, *A. cyperi*, *A. juncigena*, *A. junci-inflexi*, *A. persica*, *A. schoenoplecti*, *A. salkadehensis*, and *A. urmiana*). Detailed morphological descriptions and illustrations of these eight new species are provided, and their phylogenetic relationships with other species in the *Alternaria* section *Nimbya* are discussed below. The new host matrices for three previously known species are introduced (*A. caricicola* on *Cyperus* sp., *A. cypericola* on *Eleocharis* sp., and *A. junci-acuti* on *Carex* sp.), and a key to the recognized species in the *Alternaria* section *Nimbya* from Iran is provided.

#### 3.2.1. *Alternaria caricifolia* A. Ahmadpour, Y. Ghosta, Z. Alavi, F. Alavi and A. Poursafar, sp. nov. (Figure 4 and Figure 5) 

MycoBank No. MB 857570

Etymology. Named after the host, *Carex* sp. from which this fungus was isolated.

Typification. Iran, West Azarbaijan province, Mahabad County, isolated from the leaves and culms of *Carex* sp. (*Cyperaceae*, *Poales*) with circular to fusiform lesions, light brown to gray at center, brown to dark brown at margins, 10 July 2020, *A. Ahmadpour* (holotype IRAN 18109F, ex-type culture IRAN 4261C = FCCUU 1401).

Description. *Sexual morph* on PCA medium: *Ascomata* pseudothecial, ovoid to subglobose, dark brown to black, relatively thick-walled with flattened base, scattered or rarely aggregated, 120–240 × 90–220 μm ($\bar{x}$ = 180 × 170 μm, n = 30), formed on the surface of or embedded in culture medium. *Pseudoparaphyses* hyphoid, septate, 50–100 × 2–3 μm. *Asci* bitunicate, cylindrical to clavate, straight or slightly curved, with round apex, short pedicel, 57–80 × 15–20 μm ($\bar{x}$ = 66 × 7 μm, n = 50), 8-spored, biseriate. *Ascospores* fusiform to ellipsoidal, hyaline to pale brown, smooth, 2–4 transverse septa, 1–2 longitudinal septa and rarely with one oblique septum, conspicuously constricted at transverse septa, 17–25 × 5–10 μm ($\bar{x}$ = 21 × 7 μm, n = 50). *Asexual morph* on PCA medium: *Hyphae* branched, septate, smooth, light brown, 2–4 μm wide. *Conidiophores* macronematous, solitary, erect, simple, straight to slightly curved, septate, light brown to brown, with a single apical conidiogenous locus, or rarely 1–3 geniculate with 1–3 conidiogenous loci, 25–50 × 4–5 μm ($\bar{x}$ = 35 × 4.5 μm, n = 50). *Conidia* mostly solitary or occasionally in chains of two conidia, straight or slightly curved, obclavate to ellipsoid, conidial bodies (35–)50–87(–100) × 5–9 μm ($\bar{x}$ = 63 × 7 μm, n = 50), light brown to brown, surface smooth, (3–)6–8(–11) transverse distosepta, 1–2 eusepta, slightly constricted near eusepta, without longitudinal or oblique septum. The cell lumina are distinctly delimited and rectangular, rounded, hexagonal, or encompass the entire cell volume. True beaks are absent, but with an apical cell extension, septate, unbranched, hyaline to light brown, 12–45 × 2–3 μm, occasionally swollen at the apex. *Chlamydospores* not observed.

Culture characteristics. Colony on PCA flat, entire, floccose, white to rosy buff at center and hazel at margins, 58 mm diam after 7 days at 25 °C. Colony on PDA flat, entire, floccose, rosy buff, 50 mm diam. Colony on V-8A flat, entire, floccose, white to rosy buff, 56 mm diam. Sporulation abundant on PCA, and V-8A media, from the erect conidiophores that arise directly from the surface or the aerial hyphae. Sexual morphs were abundantly formed on PCA and V-8A after 30 days at 25 °C under fluorescent light with an 8/16 h light/dark cycle.

Additional specimen examined. Iran, West Azarbaijan province, Mahabad County, isolated from leaves and culms of *Carex* sp. (*Cyperaceae*, *Poales*), 10 July 2020, *A. Ahmadpour* (culture FCCUU 1402).

Notes. Based on the results of phylogenetic analyses (Figure 2 and Figure 3), two studied isolates of *A. caricifolia* clustered well in a separate lineage with 100% ML/MP bootstrap, and 1.0 BI posterior probabilities values, with a sister relationship to a clade consisting of *A. cyperi*, *A. salkadehensis*, and *A. schoenoplecti*. The PHI analysis confirms that *A. caricifolia* has no significant genetic recombination with closely related species (Φw = > 0.05, Figure 6). A comparison of nucleotide differences in ITS–rDNA, *GAPDH*, *TEF1*, *RPB2*, and *Alt a 1* indicates that *A. caricifolia* type strain (IRAN 4261C) differs from *A. cyperi* type strain (IRAN 4223C) by 21/456 bp (4.60%) in ITS–rDNA, 31/495 bp (6.26%, with five gaps (1%)) in *GAPDH*, 17/170 bp (10%, with four gaps (2%)) in *TEF1*, 36/746 bp (4.82%) in *RPB2* and 57/426 bp (13.38%, with two gaps (0%)) in *Alt a 1*, from *A. salkadehensis* type strain (IRAN 4225C) by 23/457 bp (5.03%, with one gap (0%)) in ITS–rDNA, 22/451 bp (4.87%, with three gaps (0%)) in *GAPDH*, 12/170 bp (7.05%, with four gaps (2%)) in *TEF1*, 37/748 bp (4.94%) in *RPB2* and 53/429 bp (12.35%, with two gaps (2%)) in *Alt a 1* and from *A. schoenoplecti* type strain (IRAN 4263C) by 20/456 bp (4.38%) in ITS–rDNA, 31/495 bp (6.26%, with five gaps (1%)) in *GAPDH*, 10/170 bp (5.88%, with four gaps (2%)) in *TEF1*, 34/748 bp (4.54%) in *RPB2* and 53/426 bp (12.44%, with two gaps (0%)) in *Alt a 1*. *Alternaria caricifolia* can be differentiated of *A. cyperi*, *A. salkadehensis* and *A. schoenoplecti* by its shorter conidiophores (25–50 μm vs. 40–110 μm, (85–)150–300 μm, 40–90 μm, respectively, narrower conidia (5–9 μm vs. 10–12 μm, 10–13 μm, and 12–15(–18) μm, respectively) and presence of sexual morph.

**Figure 4 jof-11-00225-f004:**
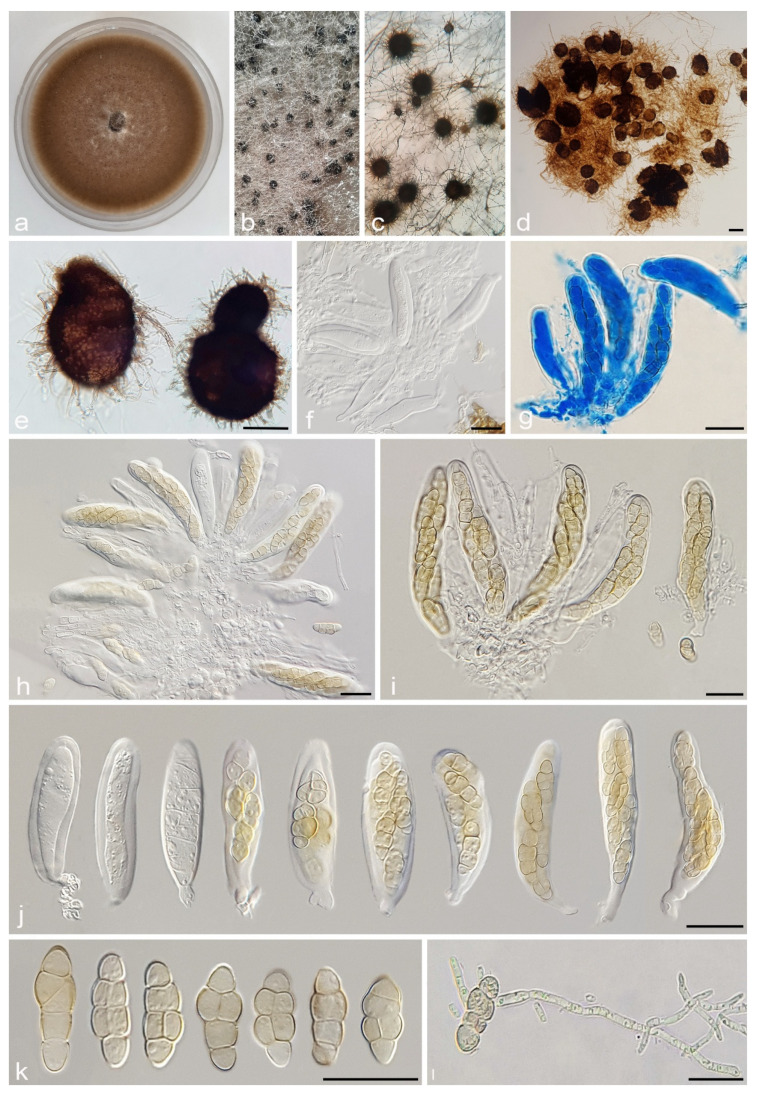
Sexual morph of *Alternaria caricifolia* (IRAN 4261C). (**a**–**e**) Ascomata formed on PCA medium after 30 days ((**b**) = 4×, (**c**) = 10×); (**f**–**j**) Asci; (**k**) Ascospores; (**l**) Germinated ascospore. Scale bars: (**d**,**e**) = 100 μm; (**f**–**l**) = 20 μm.

**Figure 5 jof-11-00225-f005:**
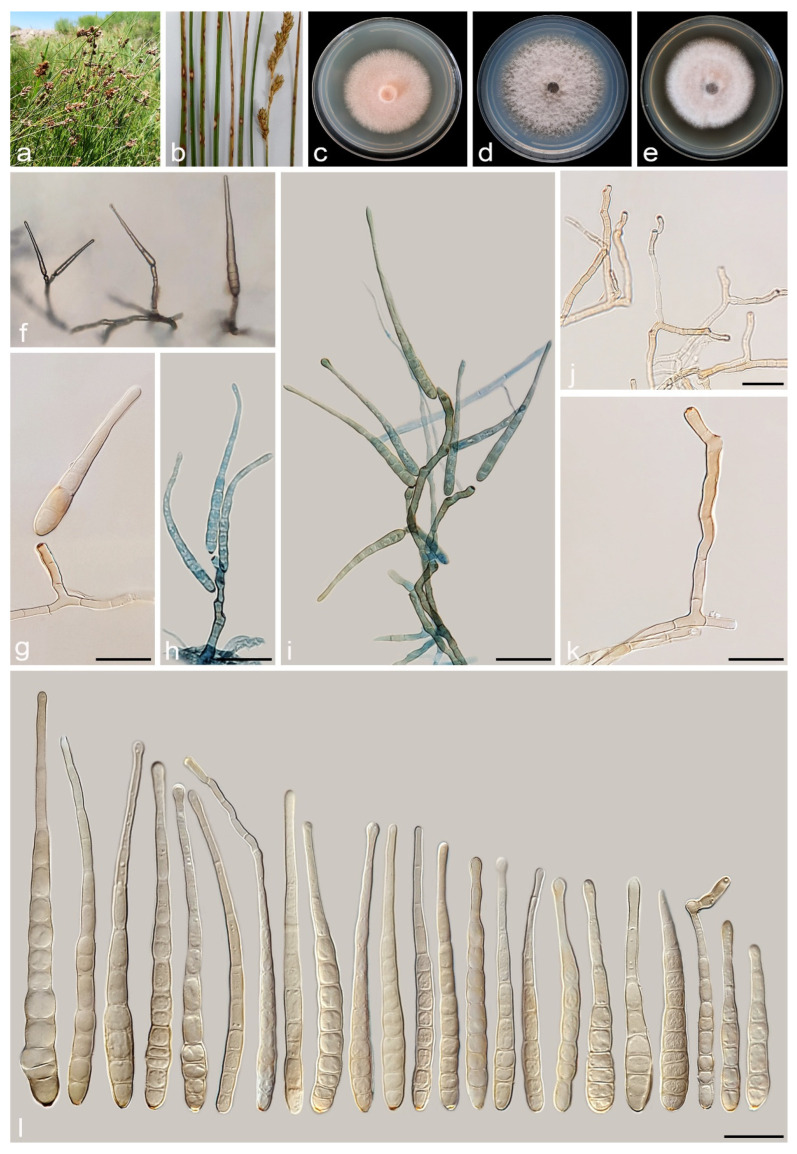
Asexual morph of *Alternaria caricifolia* (IRAN 4261C). (**a**,**b**) Symptoms on the leaves and culms of *Carex* sp.; (**c**–**e**) Colony on PDA (**c**), PCA (**d**), and V-8A (**e**) after 7 days; (**f**) Sporulation pattern on PCA (40×); (**g**–**l**) Conidiophores and conidia. Scale bars: (**g**–**l**) = 20 μm.

**Figure 6 jof-11-00225-f006:**
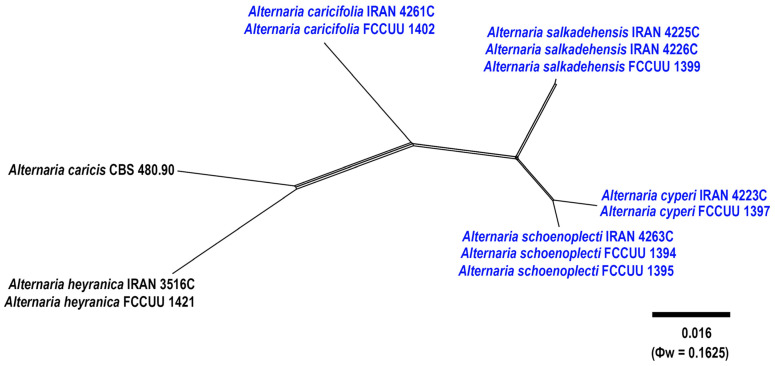
The split diagram showing the results of the pairwise homoplasy index (PHI) test of *Alternaria caricifolia* with its most closely related species (Φw = 0.1625). PHI test results (Φw) < 0.05 indicate significant recombination within the dataset. The new taxa are shown in bold blue.

#### 3.2.2. *Alternaria cyperi* A. Ahmadpour, Y. Ghosta, Z. Alavi, F. Alavi and A. Poursafar, sp. nov. (Figure 7)

MycoBank No. MB 857571

Etymology. Named after the host, *Cyperus* sp. from which this fungus was isolated.

Typification. Iran, East Azarbaijan province, Bonab County, isolated from leaves and culms of *Cyperus* sp. (*Cyperaceae*, *Poales*) with brown to black lesions and blight, 30 October 2019, *A. Ahmadpour* (holotype IRAN 18096F, ex-type culture IRAN 4223C = FCCUU 1396).

Description. *Asexual morph* on PCA medium: *Hyphae* branched, septate, smooth, light brown, 2–4 μm wide. *Conidiophores* macronematous, solitary, erect, simple, straight to slightly curved, unbranched, septate, light brown to brown, with a single apical conidiogenous locus, or 1–3 geniculate with 1–3 conidiogenous loci, 40–110 × 4–6 μm ($\bar{x}$ = 70 × 5 μm, n = 50). *Conidia* mostly in branched chains of 2–4 conidia or occasionally solitary, straight or slightly curved, obclavate to ellipsoid, apically tapered, conidial bodies (55–)60–90(–100) × 10–12 μm ($\bar{x}$ = 80 × 11 μm, n = 50), light brown to brown, surface smooth, (5–)6–9(–12) transverse distosepta, 1–5 eusepta, slightly constricted near eusepta, without longitudinal or oblique septa. True beaks are absent, but with an apical cell extension (secondary conidiophore), up to 15 µm long and 2–4 µm wide, with 1–4 geniculation and 1–4 conidiogenous loci. The cell lumina are distinctly delimited and rectangular, rounded, hexagonal, or encompass the entire cell volume. *Chlamydospores* and *sexual morph* were not observed.

Culture characteristics. Colony on PCA flat, entire, floccose, fawn with salmon aerial mycelia, 59 mm diam after 7 days at 25 °C. Colony on PDA flat, entire, floccose, white to rosy buff at center and olivaceous buff at margins, 50 mm diam. Colony on V-8A flat, entire, floccose, white at center and fulvus at margins, 25 mm diam. Sporulation is moderate to scarce on PCA, and V-8A media, from the erect conidiophores that arise directly from the surface or aerial hyphae.

Additional specimen examined. Iran, East Azarbaijan province, Bonab County, isolated from the leaves and culms of *Cyperus* sp. (*Cyperaceae*, *Poales*), 30 October 2019, *A. Ahmadpour* (culture FCCUU 1397).

Notes. *Alternaria cyperi* is phylogenetically closely related to *A. schoenoplecti* and *A. salkadehensis* (Figure 2 and Figure 3). The PHI analysis confirms that *A. cyperi* shows no significant genetic recombination with closely related species (Φw = > 0.05, Figure 6). A comparison of nucleotide differences in ITS–rDNA, *GAPDH*, *TEF1*, *RPB2*, and *Alt a 1* indicates that *A. cyperi* type strain (IRAN 4223C) differs from *A. salkadehensis* type strain (IRAN 4225C) by 2/458 bp (0.43%, with one gap (0%)) in ITS–rDNA, 12/495 bp (2.42%) in *GAPDH*, 5/165 bp (3.03%) in *TEF1*, 22/746 bp (2.94%) in *RPB2* and 43/425 bp (10.11%) in *Alt a 1* and from *A. schoenoplecti* type strain (IRAN 4263C) by 2/457 bp (0.43%) in ITS–rDNA, 8/495 bp (1.61%) in *GAPDH*, 2/165 bp (1.21%) in *TEF1*, 14/746 bp (1.87%) in *RPB2* and 10/425 bp (2.35%) in *Alt a 1*. *Alternaria cyperi* differs from *A. salkadehensis* based on shorter conidiophores (40–110 μm vs. (85–)150–300 μm) and shorter apical cell extension (up to 15 μm vs. up to 45 μm). *Alternaria schoenoplecti* morphologically differs from *A. cyperi* by having solitary and C-shaped conidia with strongly constricted at the septa.

**Figure 7 jof-11-00225-f007:**
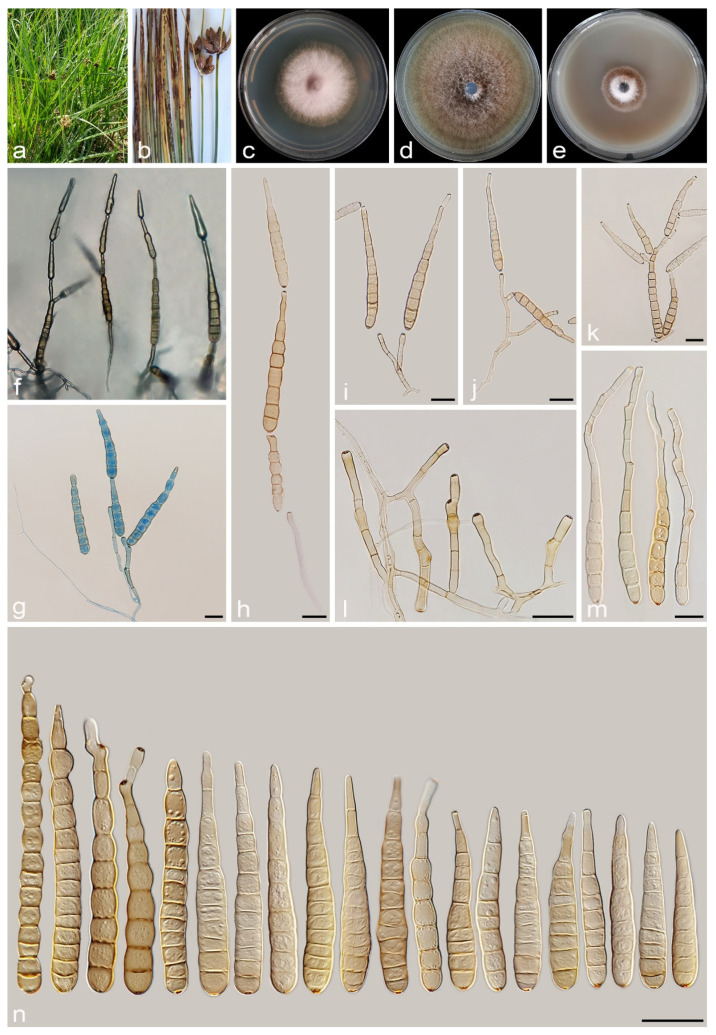
*Alternaria cyperi* (IRAN 4223C). (**a**,**b**) Symptoms on the leaves and culms of *Cyperus* sp.; (**c**–**e**) Colony on PDA (**c**), PCA (**d**), and V-8A (**e**) after 7 days; (**f**) Sporulation pattern on PCA (40×); (**g**–**n**) Conidiophores and conidia. Scale bars: (**g**–**n**) = 20 μm.

#### 3.2.3. *Alternaria juncigena* A. Ahmadpour, Y. Ghosta, Z. Alavi, F. Alavi and A. Poursafar, sp. nov. (Figure 8) 

MycoBank No. MB 857572

Etymology. Named after the host, *Juncus* sp. from which this fungus was isolated.

Typification. Iran, West Azarbaijan province, Khoy County, Salkadeh Village, isolated from the culms of *Juncus* sp. (*Juncaceae*, *Poales*), with light to dark brown irregular lesions, 25 September 2020, *A. Ahmadpour* (holotype IRAN 18211F, ex-type culture IRAN 4779C = FCCUU 1408).

Description. *Asexual morph* on PCA medium: *Hyphae* branched, septate, smooth, light brown, 2–4 μm wide. *Conidiophores* macronematous, solitary, erect, simple, septate, brown, 1–3 geniculate, with 1–3 conidiogenous loci, 40–85 × 5–6 μm ($\bar{x}$ = 57 × 5.5 μm, n = 50). *Conidia* solitary or in chains of 2–3 conidia, straight or slightly curved, obclavate to ellipsoid, conidial bodies (40–)55–90(–110) × 10–14(–16) μm ($\bar{x}$ = 70 × 12 μm, n = 50), light brown to brown, surface smooth, (5–)8–12(–15) transverse distosepta, 1–3 eusepta, slightly constricted near eusepta, with 1–3 longitudinal and 0–2 oblique septa. True beaks are absent but with apical cell extension, up to 50 µm in length and 2–3 µm wide, occasionally swollen at apex, rarely with 1–3 geniculate and 1–3 conidiogenous loci. The cell lumina are distinctly delimited and rectangular, rounded, hexagonal, or encompass the entire cell volume. *Chlamydospores* and *sexual morph* were not observed.

Culture characteristics. Colony on PCA flat, entire, velvety, pale vinaceous to vinaceous buff, with sparse aerial mycelia, 50 mm diam after 7 days at 25 °C. Colony on PDA flat, entire, floccose, white at center, and rosy buff at margins, 60 mm diam. Colony on V-8A flat, entire, floccose, white to pale violaceous, 68 mm diam. Sporulation moderate to scarce on PCA, and V-8A media, from the erect conidiophores that arise directly from the surface or the aerial hyphae.

Additional specimens examined. Iran, West Azarbaijan province, Khoy County, Salkadeh Village, isolated from the culms of *Juncus* sp., 25 September 2020, *A. Ahmadpour* (cultures FCCUU 1409, FCCUU 1410, FCCUU 1411, FCCUU 1412).

Notes. Phylogenetically, *Alternaria juncigena* forms a distinct lineage with high support values (ML/MP/BI= 97/88/0.99) (Figure 2), and is closely related to *A. junci-inflexi* and *A. persica*. A comparison of nucleotide differences in ITS–rDNA, *GAPDH*, *TEF1*, *RPB2*, and *Alt a 1* indicates that *A. juncigena* type strain (IRAN 4779C) differs from *A. junci-inflexi* type strain (IRAN 4227C) by 2/449 bp (0.44%) in ITS–rDNA, 6/490 bp (1.22%) in *GAPDH*, 2/169 bp (1.18%) in *TEF1*, 10/690 bp (1.44%) in *RPB2* and 8/428 bp (1.86%) in *Alt a 1*, and from *A. persica* type strain (IRAN 4262C) by 3/450 bp (0.66%, with one gap (0%)) in ITS–rDNA, 10/490 bp (2.04%) in *GAPDH*, 5/170 bp (2.94%, with one gap (0%)) in *TEF1*, 11/748 bp (1.47%) in *RPB2* and 2/428 bp (0.46%) in *Alt a 1*. The PHI analysis confirms that *A. juncigena* has no significant genetic recombination with closely related species (Φw = > 0.05, Figure 9). *Alternaria juncigena* differs from *A. junci-inflexi* and *A. persica* by having shorter conidiophores (40–85 μm vs. (55–)100–175(–212) μm, 45–110 μm, respectively), smooth surface, longer and wider conidia ((40–)55–90(–110) × 10–16 μm vs. smooth to verrucose surface, ((21–)45–65(–80) × (8–)9–12 μm in *A. junci-inflexi* and smooth to verrucose surface, (40–)55–85 × 8–11 μm in *A. persica*) and longer apical cell extension (up to 50 μm vs. up to 15 μm in *A. junci-inflexi*).

**Figure 8 jof-11-00225-f008:**
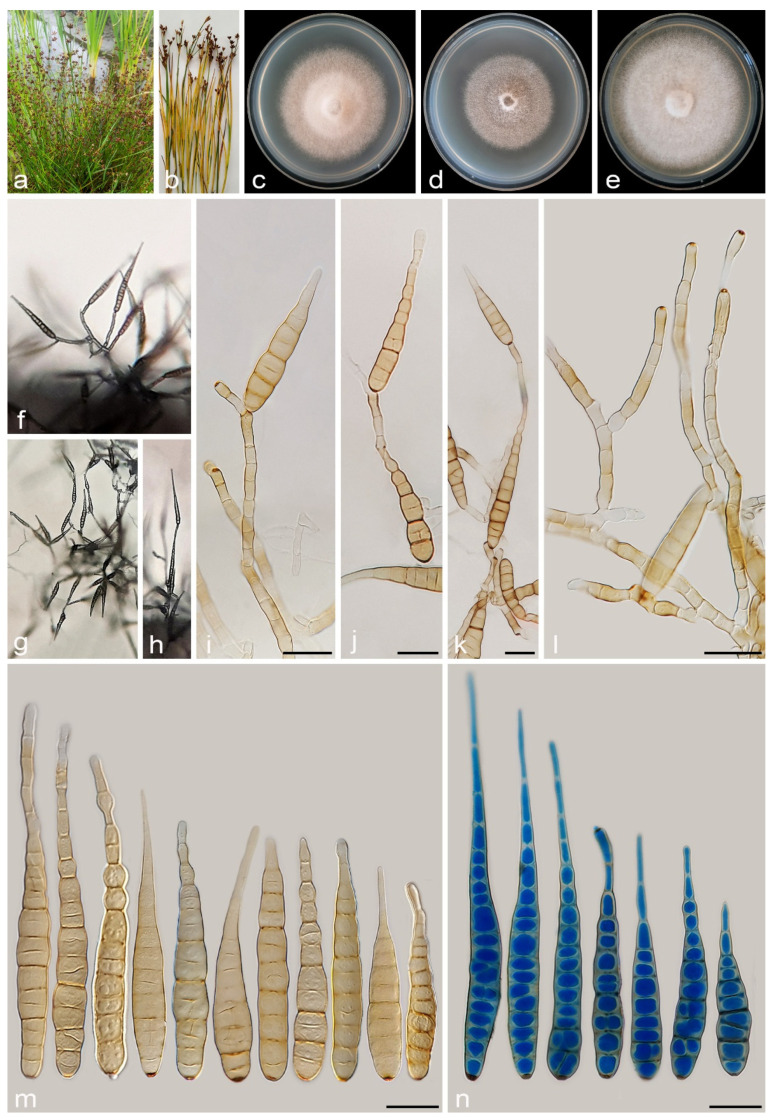
*Alternaria juncigena* (IRAN 4779C). (**a**,**b**) Symptoms on *Juncus* sp.; (**c**–**e**) Colony on PDA (**c**), PCA (**d**), and V-8A (**e**) after 7 days; (**f**–**h**) Sporulation pattern on PCA (f = 40×, g,h = 20×); (**i**–**n**) Conidiophores and conidia. Scale bars: (**i**–**n**) = 20 μm.

**Figure 9 jof-11-00225-f009:**
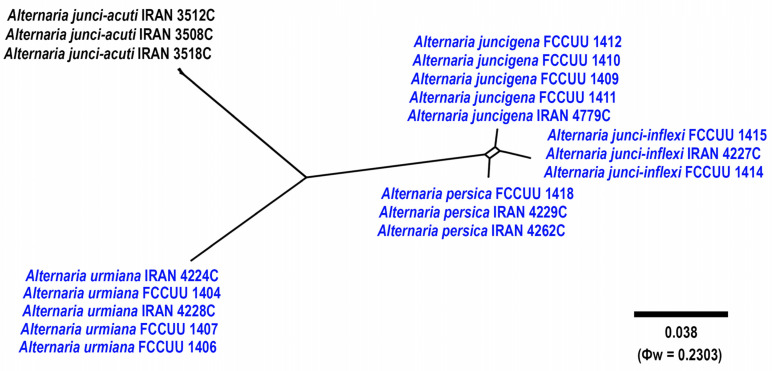
The split diagram showing the results of the pairwise homoplasy index (PHI) test of *Alternaria juncigena* with its most closely related species (Φw = 0.2303). PHI test results (Φw) < 0.05 indicate significant recombination within the dataset. The new taxa are shown in bold blue.

#### 3.2.4. *Alternaria junci-inflexi* A. Ahmadpour, Y. Ghosta, Z. Alavi, F. Alavi and A. Poursafar, sp. nov. (Figure 10)

MycoBank No. MB 857573

Etymology. Named after the host, *Juncus inflexus*, from which the fungus was isolated.

Typification. Iran, West Azarbaijan province, Khoy County, Salkadeh Village, isolated from the culms of *Juncus inflexus* (*Juncaceae*, *Poales*) with light brown to dark brown irregular lesions, 25 Sept. 2020, *A. Ahmadpour* (holotype IRAN 18099F, ex-type culture IRAN 4227C = FCCUU 1413).

Description. *Asexual morph* on PCA: *Hyphae* branched, septate, light brown, smooth, 2–4 μm wide. *Conidiophores* macronematous, solitary, straight or slightly curved, simple, unbranched, septate, light brown to brown, mostly with a single apical conidiogenous locus, or 1–5 geniculate with 1–5 conidiogenous loci, (55–)100–175(–212) × 4–5 μm ($\bar{x}$ = 135 × 4.5 μm, n = 50). *Conidia* mostly solitary, occasionally in chains of 2(–4) conidia, straight or slightly curved, mostly obclavate to narrowly ellipsoid or ovoid, conidial bodies (21–)45–65(–80) × (8–)9–12 μm ($\bar{x}$ = 51 × 10.5 μm, n = 50), light brown to brown, smooth to verrucose, 2–9 (mostly 3–7) transverse distosepta, 1–3 transverse eusepta, with 1–3 longitudinal and 0–1 oblique distosepta, and slightly constricted at eusepta. True beaks are absent but with apical cell extension, light brown to brown, unbranched, 5–15 × 2–4 μm, occasionally without apical extension. The cell lumina are distinctly delimited and rectangular, rounded, hexagonal, or encompass the entire cell volume. *Chlamydospores* and *sexual morph* were not observed. *Chlamydospores* and *sexual morph* were not observed.

Culture characteristics. Colony on PCA flat, entire, velvety, fawn with off-white aerial mycelia, reaching 65 mm diam after 7 days at 25 °C. Colony on PDA flat, entire, felty, rosy buff with white aerial mycelia, 60 mm diam. Colony on V-8A flat, entire, felty, fawn with off-white aerial mycelia, 59 mm diam. Sporulation is abundant on PCA and V-8A media, from the erect conidiophores that arise directly from the surface or the aerial hyphae.

Additional specimens examined. Iran, West Azarbaijan province, Khoy County, Salkadeh Village, isolated from culms of *Juncus inflexus*, 25 Sept. 2020, *A. Ahmadpour* (culture FCCUU 1414).—Iran, East Azarbaijan province, Maraghe County, isolated from culms of *Juncus inflexus*, 20 Sept. 2019, *A. Ahmadpour* (culture FCCUU 1415).

Notes. *Alternaria junci-inflexi* is phylogenetically (Figure 2 and Figure 3) closely related to *A. juncigena* and *A. persica*, but can be distinguished by its shorter conidia ((21–)45–65(–80) μm vs. (40–)55–90(–110) μm in *A. juncigena* and (40–)55–85 μm in *A. persica*), fewer transverse distosepta (2–9 vs. (5–)8–12(–15) in *A. juncigena* and (4–)7–12(–15) in *A. persica*) and shorter apical cell extension (5–15 μm vs. up to 50 μm in *A. juncigena* and *A. persica*). The PHI analysis confirms that *A. junci-inflexi* has no significant genetic recombination with closely related species (Φw = > 0.05, Figure 9). A comparison of nucleotide differences in ITS–rDNA, *GAPDH*, *TEF1*, *RPB2*, and *Alt a 1* indicates that *A. junci-inflexi* type strain (IRAN 4227C) differs from *A. juncigena* type strain (IRAN 4779C) by 2/449 bp (0.44%) in ITS–rDNA, 6/490 bp (1.22%) in *GAPDH*, 2/169 bp (1.18%) in *TEF1*, 10/690 bp (1.44%) in *RPB2* and 8/428 bp (1.86%) in *Alt a 1*, and from *A. persica* type strain (IRAN 4262C) by 3/450 bp (0.66%, with one gap (0%)) in ITS–rDNA, 9/490 bp (1.83%) in *GAPDH*, 7/170 bp (4.11%, with one gap (0%)) in *TEF1*, 12/690 bp (1.73%) in *RPB2* and 1/428 bp (0.23%) in *Alt a 1*.

**Figure 10 jof-11-00225-f010:**
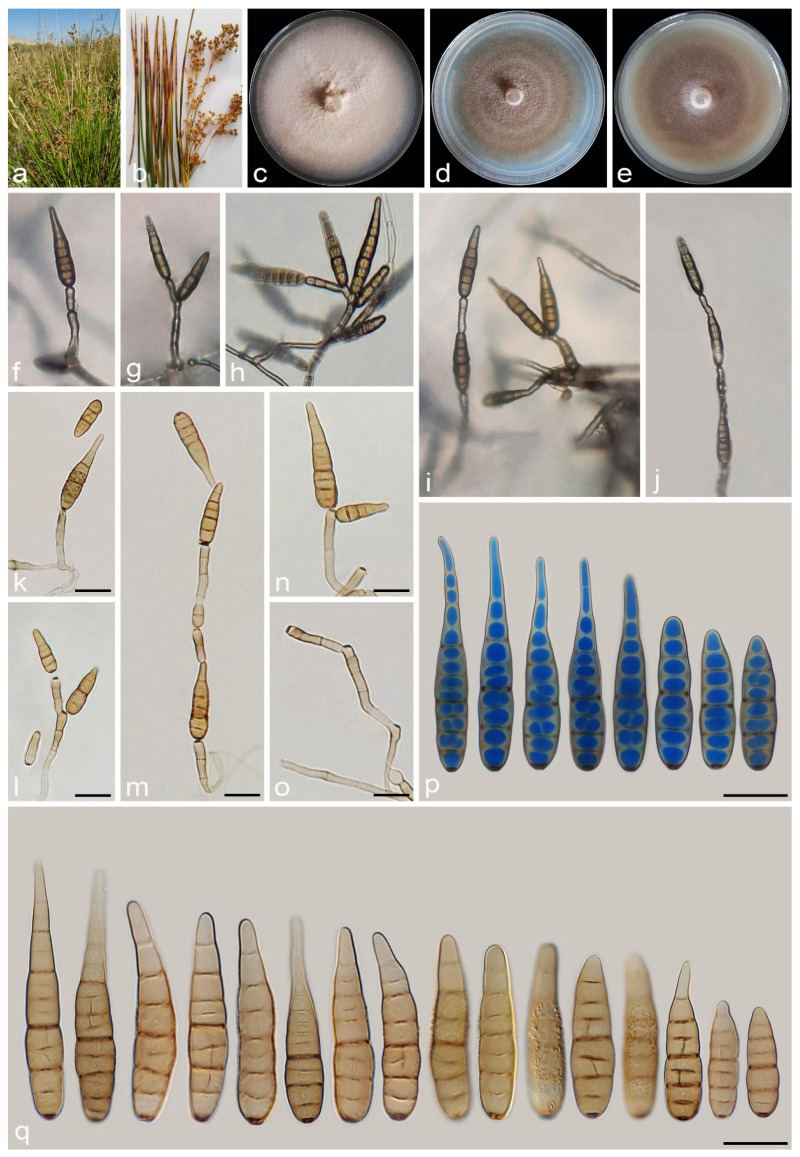
*Alternaria junci-inflexi* (IRAN 4227C). (**a**,**b**) Symptoms on culms of *Juncus inflexus*; (**c**–**e**) Colony on PDA (**c**), PCA (**d**), and V-8A (**e**) after 7 days; (**f**–**j**) Sporulation pattern on PCA (40×); (**k**–**q**) Conidiophores and conidia. Scale bars: (**k**–**q**) = 20 μm.

#### 3.2.5. *Alternaria persica* A. Ahmadpour, Y. Ghosta, Z. Alavi, F. Alavi and A. Poursafar, sp. nov. (Figure 11)

MycoBank No. MB 857574

Etymology. The name refers to the old name of Iran, Persia, where the fungus was collected.

Typification. Iran, West Azarbaijan province, Khoy County, Salkadeh Village, isolated from the culms of *Juncus* sp. (*Juncaceae*, *Poales*), with irregular brown lesions and blight, 25 September 2020, *A. Ahmadpour* (holotype IRAN 18110F, ex-type culture IRAN 4262C = FCCUU 1417).

Description. *Asexual morph* on PCA medium: *Hyphae* branched, septate, smooth, light brown, 2–4 μm wide. *Conidiophores* macronematous, solitary, erect, simple, septate, brown, 1–4 geniculate, with 1–4 conidiogenous loci, 45–110 × 4–5 μm ($\bar{x}$ = 88 × 4.5 μm, n = 50). *Conidia* solitary or in chains of 2–3 conidia, straight or slightly curved, obclavate, ellipsoid to ovoid, surface smooth to verrucose, conidial bodies (40–)55–85 × 8–11 μm ($\bar{x}$ = 65 × 9.5 μm, n = 50), light brown to brown, (4–)7–12(–15) transverse distosepta, 1–3 eusepta, slightly constricted near eusepta, with 0–3 longitudinal and 0–2 oblique septa. True beaks are absent but with apical cell extension, light brown to brown, septate, unbranched, 10–50 × 2–3 μm, occasionally swollen at the apex. The cell lumina are distinctly delimited and rectangular, rounded, hexagonal, or encompass the entire cell volume. *Chlamydospores* and *sexual morph* were not observed.

Culture characteristics. Colony on PCA flat, entire, floccose, vinaceous buff with off-white aerial mycelium, 60 mm diam after 7 days at 25 °C. Colony on PDA flat, entire, floccose, rosy buff, 64 mm diam. Colony on V-8A flat, entire, floccose, vinaceous buff with white aerial mycelium, 61 mm diam. Sporulation moderate to scarce on PCA, and V-8A media, from the erect conidiophores that arise directly from the surface or the aerial hyphae.

Additional specimens examined. Iran, West Azarbaijan province, Khoy County, Salkadeh Village, isolated from culms of *Juncus* sp., 25 September 2020, *A. Ahmadpour* (cultures IRAN 4229C = FCCUU 1416, FCCUU 1417).

Notes. Phylogenetically, *A. persica* isolates formed a monophyletic lineage with high support values (ML/MP/BI = 100/100/1.0) and is closely related to *A*. *junci-inflexi* and *A*. *juncigena* (Figure 2 and Figure 3). However, it can be distinguished from *A. junci-inflexi* based on the larger conidia ((40–)55–85 × 8–11 μm vs. (21–)45–65(–80) × (8–)9–12 in *A. juci-inflexi*), more transverse distosepta ((4–)7–12(–15) vs. 2–9 in *A. junci-inflexi*) and longer apical cell extension (10–50 μm vs. 5–15 μm in *A. junci-inflexi*) and from *A. juncigena* by having longer conidiophores (45–110 μm vs. 40–85 μm in *A. juncigena*) and smooth to verrucose surface and narrower conidia (8–11 μm vs. smooth surface, 10–16 μm in *A. juncigena*). Comparisons involving the nucleotide differences and PHI analysis (Φw = > 0.05, Figure 9) in these species are listed in the *A. juncigena* and *A. junci-inflexi* notes section.

**Figure 11 jof-11-00225-f011:**
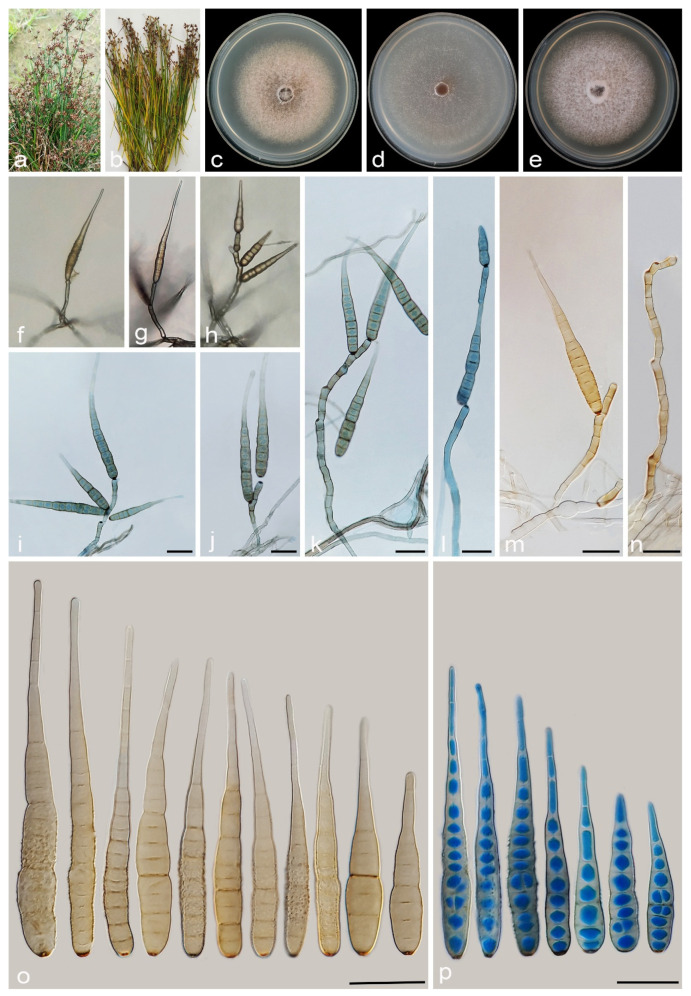
*Alternaria persica* (IRAN 4262C). (**a**,**b**) Symptoms on *Juncus* sp.; (**c**–**e**) Colony on PDA (**c**), PCA (**d**), and V-8A (**e**) after 7 days; (**f**–**h**) Sporulation pattern on PCA (40×); (**i**–**p**) Conidiophores and conidia. Scale bars: (**i**–**p**) = 20 μm.

#### 3.2.6. *Alternaria salkadehensis* A. Ahmadpour, Y. Ghosta, Z. Alavi, F. Alavi and A. Poursafar, sp. nov. (Figure 12)

MycoBank No. MB 857575

Etymology. Named after the location, Salkadeh Village, Khoy County, from where the type was collected.

Typification. Iran, West Azarbaijan province, Khoy, Salkadeh Village, from the leaves and culms of *Cyperus* sp. (*Cyperaceae*, *Poales*) with light brown to brown irregular lesions and blight, 10 Jul. 2019, A. Ahmadpour (holotype IRAN 18098F, ex-type culture IRAN 4225C = FCCUU 1400).

Description. *Asexual morph* on PCA medium: *Hyphae* branched, septate, smooth, light brown, 2–4 μm wide. *Conidiophores* macronematous, solitary, erect, simple, septate, brown, 1–5 geniculate, with 1–5 conidiogenous loci, (85–)150–300 × 5–6 μm ($\bar{x}$ = 200 × 5.5 μm, n = 50). *Conidia* mostly chains of 2–4 conidia, occasionally solitary, straight or slightly curved, obclavate, ellipsoid to ovoid, conidial bodies (35–)55–90(–100) × 10–13 μm ($\bar{x}$ = 70 × 11 μm, n = 50), light brown to brown, surface smooth, (4–)5–9(–13) transverse distosepta, 1–3 eusepta, slightly constricted near eusepta, without longitudinal or oblique septa. True beaks are absent, but with apical cell extension (secondary conidiophores), up to 45 µm long and 2–4 µm wide, with 3–4(–6) geniculate and 3–4(–6) conidiogenous loci, each bearing one conidium. The cell lumina are distinctly delimited and rectangular, rounded, hexagonal, or encompass the entire cell volume. *Chlamydospores* and *sexual morph* were not observed.

Culture characteristics. Colony on PCA flat, entire, velvety, hazel, with sparse aerial mycelium, 63 mm diam after 7 days at 25 °C. Colony on PDA flat, entire, floccose, rose buff with white aerial mycelium, 55 mm diam. Colony on V-8A flat, entire, velvety, dark brick at the center and ochreous at the margin, with sparse off-white aerial mycelium, 63 mm diam. Sporulation is abundant on PCA and V-8A media, from the erect conidiophores that arise directly from the surface or aerial hyphae.

Additional specimens examined. Iran, West Azarbaijan province, Khoy County, Salkadeh Village, from the leaves and culms of *Cyperus* sp. (*Cyperaceae*, *Poales*), 10 July 2019, A. Ahmadpour (culture FCCUU 1399).—*ibid.* on leaves and culms of *Carex* sp. (*Cyperaceae*, *Poales*), 10 July 2019, A. Ahmadpour (culture IRAN 4226C = FCCUU 1398).

Notes. *Alternaria salkadehensis* is phylogenetically closely related to *A. cyperi* and *A. schoenoplecti* (ML/MP/BI= 100/100/1.0) (Figure 2 and Figure 3), but it can be differentiated from *A. cyperi* by its longer conidiophores (85–)150–300 μm vs. 40–110 μm) and longer apical cell extension (up to 45 μm vs. up to 15 μm). *Alternaria schoenoplecti* morphologically differs from *A. salkadehensis* by having solitary and straight or curved conidia (C shape) vs. catenate, straight or slightly curved conidia. The PHI analysis confirms that *A. salkadehensis* has no significant genetic recombination with closely related species (Φw = > 0.05, Figure 6). A comparison of nucleotide differences in ITS–rDNA, *GAPDH*, *TEF1*, *RPB2*, and *Alt a 1* indicates that *A. salkadehensis* type strain (IRAN 4225C) differs from *A. cyperi* type strain (IRAN 4223C) by 2/458 bp (0.43%, with one gap (0%)) in ITS–rDNA, 12/495 bp (2.42%) in *GAPDH*, 5/165 bp (3.03%) in *TEF1*, 22/746 bp (2.94%) in *RPB2* and 43/425 bp (10.11%) in *Alt a 1* and from *A. schoenoplecti* type strain (IRAN 4263C) by 4/458 bp (0.87%, with one gap (0%)) in ITS–rDNA, 12/495 bp (2.42%) in *GAPDH*, 4/167 bp (2.39%) in *TEF1*, 23/748 bp (3.07%) in *RPB2* and 39/428 bp (9.11%) in *Alt a 1*.

**Figure 12 jof-11-00225-f012:**
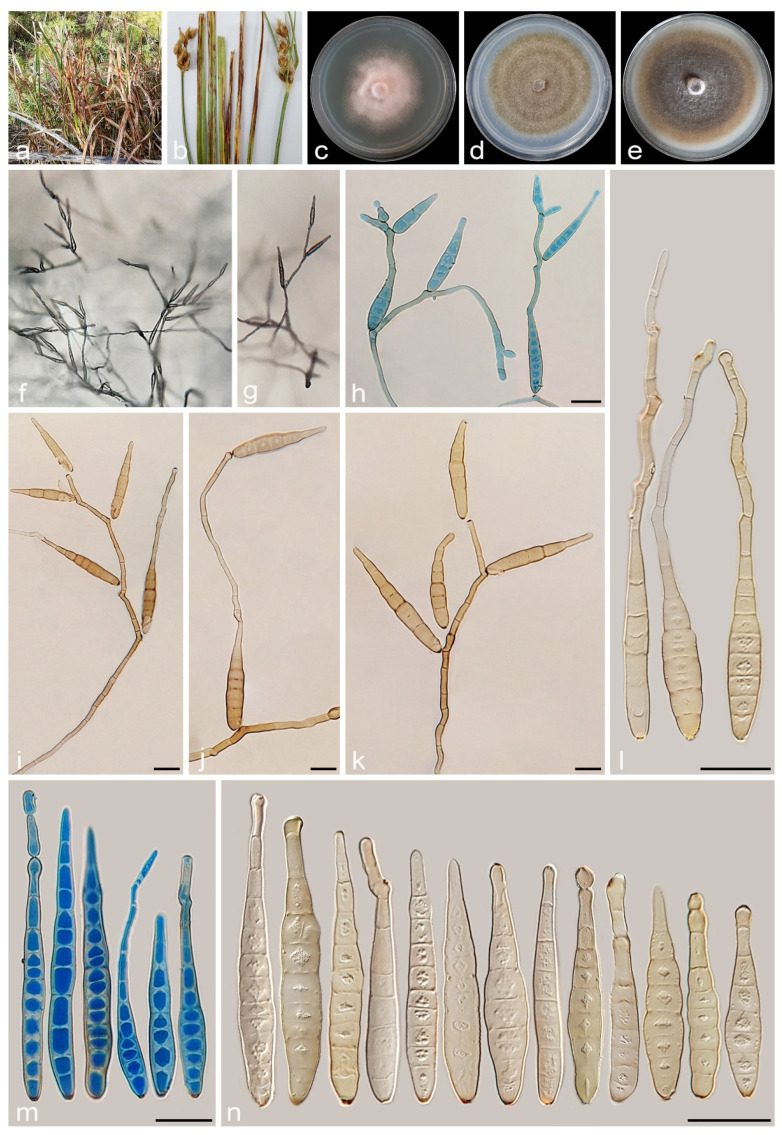
*Alternaria salkadehensis* (IRAN 4225C). (**a**,**b**) Symptoms on the leaves and culms of *Cyperus* sp.; (**c**–**e**) Colony on PDA (**c**), PCA (**d**), and V-8A (**e**) after 7 days; (**f**,**g**) Sporulation pattern on PCA (20×); (**h**–**n**) Conidiophores and conidia. Scale bars: (**h**–**n**) = 20 μm.

#### 3.2.7. *Alternaria schoenoplecti* A. Ahmadpour, Y. Ghosta, Z. Alavi, F. Alavi and A. Poursafar, sp. nov. (Figure 13)

MycoBank No. MB 857576

Etymology. Named after the host, *Schoenoplectus* sp. from which the fungus was isolated.

Typification. Iran, West Azarbaijan province, Miyandoab County, from the culms of *Schoenoplectus* sp. (*Cyperaceae*, *Poales*), with brown to dark brown, ovoid to fusiform lesions, 10 June 2019, *A. Ahmadpour* (holotype IRAN 18111F, ex-type culture IRAN 4263C = FCCUU 1393).

Description. *Asexual morph* on PCA medium: *Hyphae* branched, septate, smooth, light brown, 2–4 μm wide. *Conidiophores* macronematous, solitary, erect, simple, straight to slightly curved, septate, light brown to brown, with a single apical conidiogenous locus, 40–90 × 5–7 μm ($\bar{x}$ = 66 × 6 μm, n = 50). *Conidia* mostly solitary, rarely in chains of two conidia, straight or curved, obclavate to narrowly ellipsoid, conidial bodies (40–)50–70(–88) × 12–15(–18) μm ($\bar{x}$ = 63 × 14 μm, n = 50), light brown to brown, smooth, (3–)4–8(–11) transverse distosepta, 1–3 eusepta, rarely with 1–2 longitudinal or oblique distosepta. Mature conidia are strongly constricted at most of their transverse septa, and in some conidia, the median cells are markedly swollen. True beaks are absent, but with an apical cell extension up to 22 μm long and 3–5 μm wide, occasionally swollen at the apex. The cell lumina are distinctly delimited and rectangular, rounded, hexagonal, or encompass the entire cell volume. *Chlamydospores* and *sexual morph* were not observed.

Culture characteristics. Colony on PCA flat, entire, velvety, sepia to cinnamon, with sparse aerial hyphae, 49 mm diam. after 7 days at 25 °C. Colony on PDA flat, entire, floccose, dark brick to cinnamon, 55 mm diam. Colony on V-8A flat, entire, velvety, sepia with white aerial mycelia, 30 mm diam. Sporulation is scarce on PCA, V-8A, or SNA media, from the erect conidiophores that arise directly from the surface or aerial hyphae.

Additional specimens examined. Iran, West Azarbaijan province, Miyandoab County, from culms of *Schoenoplectus* sp., 10 June 2019, *A. Ahmadpour* (cultures FCCUU 1394, FCCUU 1395).

Notes. *Alternaria schoenoplecti* is phylogenetically closely related to *A. cyperi* and *A. salkadehensis* (Figure 2 and Figure 3). Comparisons of nucleotide differences and PHI analysis (Φw = > 0.05, Figure 6) in these species are listed in the *A. cyperi* and *A. salkadehensis* notes section. Moreover, *A. schoenoplecti* differs from *A. cyperi* and *A. salkadehensis* morphologically in having solitary and straight or curved conidia (C shape) vs. catenate, straight or slightly curved conidia.

**Figure 13 jof-11-00225-f013:**
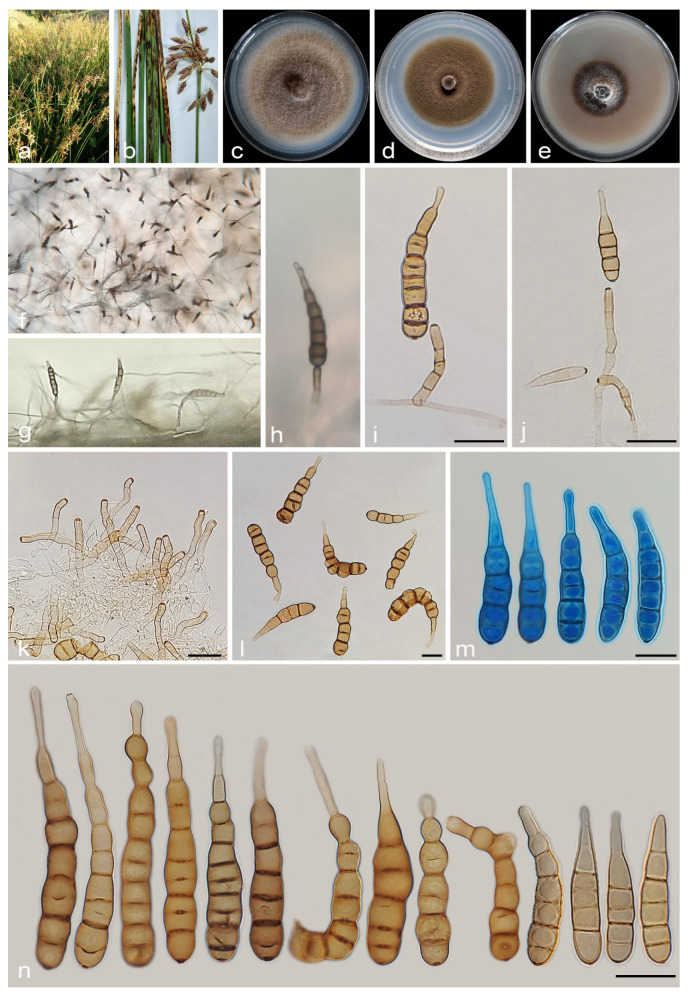
*Alternaria schoenoplecti* (IRAN 4263C). (**a**,**b**) Symptoms on culms of *Schoenoplectus* sp.; (**c**–**e**) Colony on PDA (**c**), PCA (**d**), and V-8A (**e**) after 7 days; (**f**–**h**) Sporulation pattern on PCA (f = 10×, g = 20×, h = 40×); (**i**–**n**) Conidiophores and conidia. Scale bars: (**i**–**n**) = 20 μm.

#### 3.2.8. *Alternaria urmiana* A. Ahmadpour, Y. Ghosta, Z. Alavi, F. Alavi and A. Poursafar, sp. nov. (Figure 14)

MycoBank No. MB 857577

Etymology. The name refers to the Urmia County, where the holotype was collected.

Typification. Iran, West Azarbaijan province, Urmia County, isolated from the culms of *Juncus acutus* (*Juncaceae*, *Poales*) with brown lesions and blight symptoms, 20 September 2019, *Y. Ghosta* (holotype IRAN 18097F, ex-type culture IRAN 4224C = FCCUU 1405).

Description. *Asexual morph* on PCA: *Hyphae* branched, septate, light brown, smooth, 2–4 μm wide. *Conidiophores* macronematous, solitary, straight or slightly curved, simple, unbranched, septate, light brown to brown, mostly with a single apical conidiogenous locus, or 1–4 geniculate with 1–4 conidiogenous loci, (75–)100–150(–225) × 4–5 μm ($\bar{x}$ = 136 × 4.5 μm, n = 50). *Conidia* mostly solitary, rarely in chains of two conidia, straight or slightly curved, mostly obclavate, ellipsoid to ovoid, conidial bodies (35–)40–51(–55) × 12–15 μm ($\bar{x}$ = 46 × 13 μm, n = 50), light brown to brown, smooth to verrucose, 3–8 (mostly 3–6) transverse distosepta, 1–2 transverse eusepta, with 1–3 longitudinal and 0–2 oblique distosepta. Conidium body slightly constricted at eusepta, mostly without apical cell extension, occasionally with apical cell extension, light brown to brown, 4–10 × 2–4 μm. The cell lumina are distinctly delimited and rectangular, rounded, hexagonal, or encompass the entire cell volume. *Chlamydospores* and *sexual morph* were not observed.

Culture characteristics. Colony on PCA flat, entire, velvety, rosy buff with sparse, off-white aerial mycelium, reaching 51 mm diam. after 7 days at 25 °C. Colony on PDA flat, entire, floccose, dark birk to sepia at the center and white at the margin, with sparse aerial mycelium, 50 mm diam. Colony on V-8A flat, entire, floccose, white at center, fawn at margins, 25 mm diam. Sporulation abundant on PCA, and V-8A media, from the erect conidiophores that arise directly from the surface or the aerial hyphae.

Additional specimens examined. Iran, West Azarbaijan province, Urmia County, isolated from culms of *Juncus acutus*, 20 September 2019, *Y. Ghosta* (cultures FCCUU 1406, FCCUU 1407).—Iran, West Azarbaijan, Khoy County, Salkadeh Village, isolated from culms of *Juncus inflexus*, 25 September 2020, *A. Ahmadpour* (cultures IRAN 4228C, FCCUU 1404).

Notes. Phylogenetically, *Alternaria urmiana* clusters in a distinct subclade compared to all other species in *Alternaria* section *Nimbya*, with 100% ML/MP bootstrap, and 1.0 BI posterior probability values, with a sister relationship to a clade consisting *A. junci-acuti* (Figure 2 and Figure 3). A comparison of nucleotide differences in ITS–rDNA, *GAPDH*, *TEF1*, *RPB2*, and *Alt a 1* indicates that *A. urmiana* type strain (IRAN 4224C) differs from *A. junci-acuti* type strain (IRAN 3512C) by 25/433 bp (5.77%, with three gaps (0%)) in ITS–rDNA, 30/490 bp (6.12%, with five gaps (1%)) in *GAPDH*, 17/157 bp (10.82%, with two gaps (1%)) in *TEF1*, 78/746 bp (10.45%) in *RPB2* and 61/428 bp (14.25%, with two gaps (0%)) in *Alt a 1*. Morphologically, *A. urmiana* is similar to *A. junci-inflexi* but can be distinguished by its smaller and wider conidia ((35–)40–51(–55) × 12–15 μm vs. (21–)45–65(–80) × (8–)9–12 μm) with shorter apical cell extension (up to 10 μm vs. up to 15 μm).

**Figure 14 jof-11-00225-f014:**
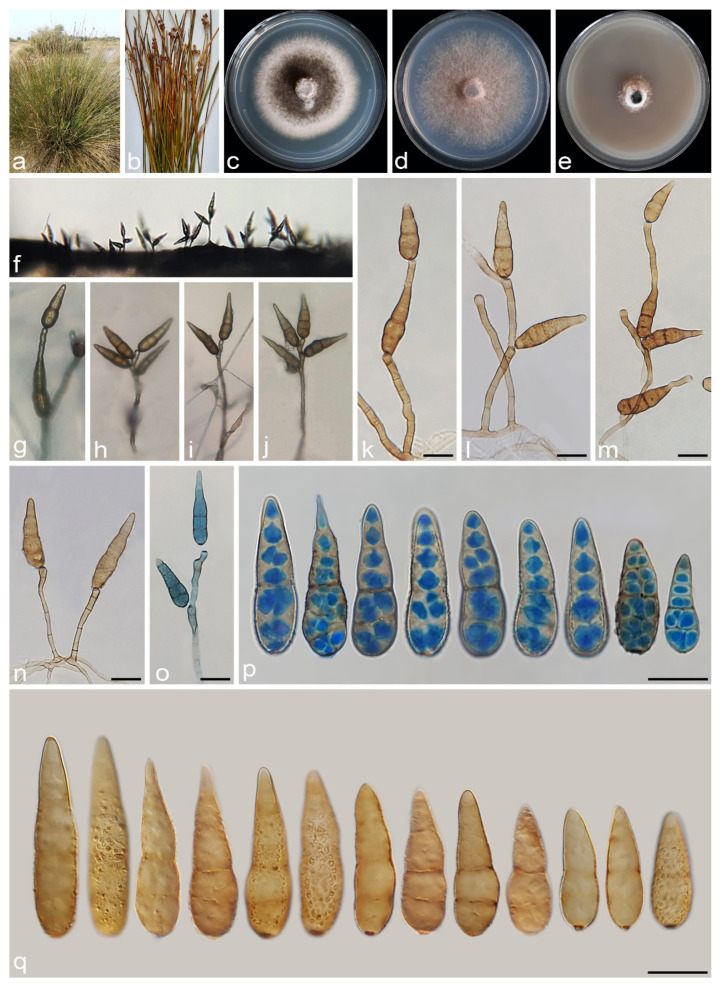
*Alternaria urmiana* (IRAN 4224C). (**a**,**b**) Symptoms on culms of *Juncus acutus*; (**c**–**e**) Colony on PDA (**c**), PCA (**d**), and V-8A (**e**) after 7 days; (**f**–**j**) Sporulation pattern on PCA (f = 10×, g–j = 40×); (**k**–**q**) Conidiophores and conidia. Scale bars: (**k**–**q**) = 20 μm.

#### 3.2.9. *Alternaria caricicola* Ahmadpour, Phytotaxa 405(2): 69. 2019 [19]

Description and Illustration. See Ahmadpour [19] and Ahmadpour et al. [20].

Habitat and Distribution. On culms of *Carex* sp. [19,20] and *Cyperus* sp. (*Cyperaceae*, *Poales*) (this study), Iran.

Materials examined. Iran, West Azarbaijan province, Mahabad County, on infected leaves and culms of *Cyperus* sp. (*Cyperaceae*, *Poales*), 10 July 2020, *A. Ahmadpour* (cultures FCCUU 1384, FCCUU 1385).—Iran, East Azarbaijan province, Bonab County, isolated from the leaves and culms of *Cyperus* sp., 30 October 2019, *A. Ahmadpour* (culture FCCUU 1386).

Notes. *Alternaria caricicola* was originally reported from the culms of *Carex* sp. with gray to brown lesions in West Azarbaijan province, Iran [19]. In this study, three isolates from *Cyperus* sp. (FCCUU 1384, FCCUU 1385, and FCCUU 1386) were identified as *A. caricicola* in a well-supported clade with ML/MP/BI = 100/100/1.0 (Figure 2 and Figure 3), so *Cyperus* spp. is reported as *matrix nova* for this species. Our isolates (FCCUU 1384, FCCUU 1385, and FCCUU 1386) had 100% sequence homology with *A. caricicola* type strain (IRAN 3418C), according to the gene regions (ITS–rDNA, *GAPDH*, *TEF1*, *RPB2*, and *Alt a 1* genes). *Alternaria caricicola* is morphologically similar to *A. cypericola*, *A. heteroschemos*, *A. juncicola*, and *A. scirpicola* [1,9,19,20], but it can be differentiated from *A. scirpicola* and *A. heteroschemos* by the longer and narrower conidia (87.5–205 × 10–15 µm vs. 75–100 × 18–22 μm in *A. heteroschemos* and 100–120 × 15–20 μm in *A. scirpicola*) and more distosepta (8–15 vs. 6–9 in *A. heteroschemos* and 9–11 in *A. scirpicola*). *Alternaria juncicola* has shorter conidiophores (60–130 μm), more distosepta (up to 17 vs. up to 14), and eusepta (3–8 vs. 1–4) in *A. caricicola* [1,9,19,20]. *Alternaria caricicola* can be differentiated from *A. cypericola* based on longer conidia (87.5–205 µm vs. 87.5–155 µm in *A. cypericola*) and more transverse distosepta (7–14 vs. 6–12 in *A. cypericola*) [19,20].

#### 3.2.10. *Alternaria cypericola* Ahmadpour, Poursafar and Ghosta, Mycologia 113: 1078. 2021 [20]

Description and Illustration. Ahmadpour et al. [20].

Habitat and Distribution. On infected leaves and culms of *Cyperus* sp. (*Cyperaceae*) [20], *Cyperus* sp. and *Eleocharis* sp. (*Cyperaceae*, *Poales*) (this study), Iran.

Materials examined. Iran, West Azarbaijan province, Khoy County, Salkadeh Village, from the leaves and culms of *Cyperus* sp. (*Cyperaceae*, *Poales*), 10 Aug. 2020, *A. Ahmadpour* (culture FCCUU 1387).—*ibid.* on the culms of *Eleocharis* sp. (*Cyperaceae*, *Poales*), 10 Aug. 2020, *A. Ahmadpour* (culture FCCUU 1388).

Notes. *Alternaria cypericola* was first reported from infected leaves and culms of *Cyperus* sp. (*Cyperaceae*) with light brown to brown lesions in Iran [20]. In this study, two isolates (FCCUU 1387 and FCCUU 1388) obtained from the leaves and culms of *Cyperus* sp. and *Eleocharis* sp. were identified as *A. cypericola*; thus, *Eleocharis* sp. is reported here as *matrix nova* for this species. Our isolates (FCCUU 1387 and FCCUU 1388) exhibited 100% sequence homology with the *A. cypericola* type strain (IRAN 3423C) in the ITS–rDNA, *RPB2*, and *Alt a 1* genes. This species is phylogenetically closely related to, but distinct from *A. scirpicola*, *A. scirpinfestans*, and *A. scirpivora* (Figure 2 and Figure 3). *Alternaria cypericola* differs from *A. scirpicola* on the basis of longer and narrower conidia (87.5–155 × 10–15 μm vs. 100–120 × 15–20 μm in *A. scirpicola*) [1,9]. Also, it differs from *A. scirpinfestans* and *A. scirpivora* based on longer and wider conidia (90–130 × 6–8 μm in *A. scirpinfestans* and 90–120 × 7–12 μm in *A. scirpivora*) and more transverse distosepta (6–12 vs. 3–8 in *A. scirpinfestans* and 4–9 in *A. scirpivora*) [8,9].

#### 3.2.11. *Alternaria heyranica* Ahmadpour, Poursafar and Ghosta, Mycologia 113: 1078. 2021 [20]

Description and Illustration. Ahmadpour et al. [20].

Material examined. Iran, Guilan province, Astara, Heyran Village, on infected leaves of *Carex* sp. (*Cyperaceae*, *Poales*), 10 October 2020, *A. Ahmadpour* (FCCUU 1421).

Habitat and Distribution. On infected leaves of *Carex* sp. from Iran ([9], this study).

Notes. *Alternaria heyranica* was first described on infected leaves of *Carex* sp. in Guilan province, Iran. In the phylogenetic tree, one studied isolate (FCCUU 1421) clustered with the ex-type of *A. heyranica* (IRAN 3516C) with 1.0 BI posterior probabilities and 100% ML/MP bootstrap values (Figure 1 and Figure 2). The isolate FCCUU 1421 had 100% sequence homology with *A. heyranica* type strain (IRAN 3516C), according to the gene regions (ITS–rDNA, *GAPDH*, *TEF1*, *RPB2*, and *Alt a 1* genes). *Alternaria heyranica* is most closely related to *A. caricis* strain CBS 480.90, but is distinguished by having longer filiform true beaks (50–175 μm length), smaller conidia (50–75 × 10–12 μm vs. 65–95 × 10–16 μm in *A. caricis*) and formation of hyphal swelling [1,20].

#### 3.2.12. *Alternaria junci-acuti* Ahmadpour, Poursafar and Ghosta, Mycologia 113: 1080. 2021 [20]

Description and Illustration. See Ahmadpour et al. [20].

Materials examined. Iran, West Azarbaijan province, Khoy County, on infected leaves and culms of *Carex* sp. (*Cyperaceae*, *Poales*), 20 October 2019, *A. Ahmadpour* (cultures FCCUU 1390, FCCUU 1392).—Iran, West Azarbaijan province, Miyandoab County, from the leaves and culms of *Carex* sp., 30 October 2019, *A. Ahmadpour* (culture FCCUU 1391).—Iran, Golestan province, Gorgarn County, from the leaves and culms of *Carex* sp., 10 July 2020, *A. Ahmadpour* (culture FCCUU 1389).

Habitat and Distribution. On infected leaves and culms of *Juncus acutus* [20] and *Carex* sp. (this study) from Iran.

Notes. *Alternaria junci-acuti* was first identified on symptomatic culms of *Juncus acutus* in West Azarbaijan province, Urmia County, Iran [20]. In this study, four isolates of this species were recovered from the leaves and culms of *Carex* sp., leading to the identification of *Carex* sp. as a new host (*matrix nova*) for *A. Junci-acuti*. Additionally, this study confirmed the presence of this species in various geographic regions across Iran, thereby expanding its known distribution range. Our isolates (FCCUU 1389, FCCUU 1390, FCCUU 1391, and FCCUU 1392) exhibited 99–100% sequence homology with the *A. junci-acuti* type strain (IRAN 3512C) based on the gene regions (ITS–rDNA, *GAPDH*, *RPB2*, and *Alt a 1* genes). *Alternaria junci-acuti* is phylogenetically related to *A. urmiana* and forms a well-supported clade (ML/MP/BI= 100/100/1.0) (Figure 2 and Figure 3). It is morphologically similar to *A. heyranica* due to the formation of a long filiform true beak, but it can be distinguished based on the longer and narrower conidia in *A. junci-acuti* (50–87.5 × 7–9 μm) vs. *A. heyranica* (50–75 × 10–12 μm), and more transverse distosepta in conidia of *A. junci-acuti* (up to 14 vs. up to 9 in *A. heyranica*) [20]. Also, *A. junci-acuti* is readily distinguished from *A. caricis* concerning having a long filiform true beak, narrower conidia (10–16 μm in *A. caricis*), and more distosepta (up to 10 in *A. caricis*) [1,9].

#### 3.2.13. *Alternaria scirpivora* (E.G. Simmons and D.A. Johnson) Woudenb. and Crous, Stud. Mycol. 75: 198. 2013 [17]

Basionym. *Nimbya scirpivora* E.G. Simmons and D.A. Johnson, Mycotaxon 84: 424. 2002 [8].

Synonym. *Macrospora scirpivora* E.G. Simmons and D.A. Johnson, Mycotaxon 84: 422. 2002 [8].

Description and Illustration. See Johnson et al. [8], Ahmadpour et al. [20], and Alavi et al. [55].

Habitat and Distribution. On infected culms of *Scirpus acutus* and *S. validus* (*Cyperaceae*) in the USA and on *S. acutus* in Iran [8,9,20,55,56].

Materials examined. Iran, Ardebil province, Hir County, Abbas Abad Village, on infected culms of *Scirpus acutus* (*Cyperaceae*), 10 July 2020, *A. Ahmadpour* (culture FCCUU 1419).—Iran, Mazandaran province, Larim County, on infected culms of *S. acutus*, 10 October 2021, *A. Ahmadpour* (culture FCCUU 1420).

Notes. This species was originally described from culm lesions of *S. acutus* in the Pacific Northwest and Minnesota, USA [8]. In Iran, it has been reported from several locations in West Azarbaijan province [20,55], and this study further extends its known range to include Ardebil and Mazandaran provinces, broadening its geographic distribution. Our isolates (FCCUU 1419 and FCCUU 1420) had 99–100% sequence homology with *A. scirpivora* type strain (EGS 50-021), according to the gene regions (ITS–rDNA, *GAPDH*, and *Alt a 1* genes). *Alternaria scirpivora* is phylogenetically related to *A. scirpicola* and *A. scirpinfestans* (Figure 2 and Figure 3), but is distinguished concerning the sporulation pattern, size of conidia, and the number of transverse pseudosepta [8,20]. According to the phylogenetic tree, the two studied isolates were clustered well with *A. scirpivora* in a distinct subclade with 1.0 BI posterior probabilities and 100% ML/MP bootstrap values (Figure 2 and Figure 3). *Alternaria scirpivora* and *A. scirpinfestans* are pathogens of *Scirpus* spp. (*S. acutus* and *S. validus*) [8,20,55].


**A key to recognized species in the *Alternaria* section *Nimbya* from Iran**


1 Conidia with true beaks ………………………………………………………………………………………………………… 21′ Conidia without true beaks, but with apical cell extension ………………………………………………………………… 52 Beak long, more than 100 µm, chlamydospores present ……………………….……….…………………………………… 32′ Beak short, less than 100 µm, chlamydospores absent …………….………………………………………………………… 43 Chlamydospores bulbous, hyaline to light brown …………………….….….………………………….………… *A. heyranica*3′ Chlamydospores not bulbous, brown to dark brown ……………………….………………………………….. *A. junci-acuti*4 Conidia 87.5–205 × 10–15 μm, 7–14 distosepta ……………………….……………………………………………. *A. caricicola*4′ Conidia 87.5–155 × 10–15 μm, 6–12 distosepta ……………….………………………….……….………………. *A. cypericola*5 Conidia strongly curved, C shaped, on *Schoenoplectus* ………….……….……………….…………………… *A. schoenoplecti*5′ Conidia straight or slightly curved, not C shape………………………………….………………………………………….. 66 Secondary conidiophores long with 2–5 geniculations ………………………………….…………………………………… 76′ Secondary conidiophores short, without geniculations ………………….…………………………………………………. 87 Primary conidiophores long, up to 300 µm, longer apical cell extension (up to 45 μm) ……………………………………………………………………….………………………………………… *A. salkadehensis*7′ Primary conidiophores short, up to 110 µm, shorter apical cell extension (up to 15 μm) ……………………………………………………………….………………………………………………………… *A. cyperi*8 Conidia mostly solitary, rarely in chains of 2 conidia ………………………………………………………………………… 98′ Conidia solitary or mostly in chains of 2–4(–8) conidia ………………….…….…………………………………………… 109 Conidium body short and wide (35–55 × 12–15 µm), conidial surface smooth or verrucose, without ascomata ………………………………………………………………………………………….……………………… *A. urmiana*9′ Conidium body long and narrow (35–100 × 5–9 µm), conidial surface smooth, without longisepta and with ascomata ……………………………………………………………………………………….…………….………… *A. caricifolia*9″ Conidium body moderate size (21–80 × 8–12 µm), conidial surface smooth or verrucose, with 1–3 longisepta, without ascomata ……………………………………………….…….……………………………………………………… *A. junci-inflexi*10 Conidia mostly in chains of 2–8, conidium body small (20–60 × 5–8 µm) ……………………………………. *A. scirpivora*10′ Conidia mostly in chains of 2–3, conidium body medium (55–85 × 8–11 µm), conidial surface smooth or verrucose ………………………………………………………………….……………….……….…………….……… *A. persica*10″ Conidia mostly in chains of 2–4, conidium body large (40–110 × 10–16 µm), conidial surface smooth ……………………………………………………………………….………………………………………… *A. junicigena*

## 4. Discussion

This study recovered 189 fungal isolates with conidial characteristics typical of *Alternaria* section *Nimbya* from plant species in the families *Cyperaceae* and *Juncaceae*. The plant samples exhibiting blight, leaf, and culm lesions were collected from wetlands across six provinces in Iran (Table 4). The ISSR marker banding pattern effectively grouped the isolates into distinct categories, and a comparison of their morphological characteristics also aligned with the ISSR results. This finding supports our previous study, which demonstrated that ISSR markers can be used to group the isolates, with different banding patterns indicating distinct species [20]. Since evaluating morphological characteristics for a large number of isolates can be time-consuming and requires standardized and controlled cultural conditions (e.g., temperature, light, and growth medium), the ISSR marker offers a more efficient method, reducing the time required for analysis while providing clear species differentiation.

Species in the *Alternaria* section *Nimbya* belong to one of the seven *Alternaria* sections with reported sexual forms [8,17,57,58]. Lucas and Webster [59] successfully obtained the ascomata of *A. scirpicola* (≡*Macrospora scirpicola*) in mono-ascospore and mono-conidia cultures on sterilized *Cyperus* stems. Johnson et al. [8] also induced ascomata formation in *A. scirpinfestans* and *A. scirpivora* by inoculating autoclave-sterilized culm sections of host plants on 2% water agar/PCA, with ascomata maturing within three to four weeks. Notably, *A. scirpivora* exhibited faster maturation than *A. scirpinfestans*, confirming its homothallic nature. Despite attempts to induce ascomata formation in the isolates studied in this work, it was only observed in *A. caricifolia*, suggesting its homothallic nature. Although *Alternaria* has traditionally been considered an asexual genus, several species possess functional mating type genes (*MAT* loci) that resemble those of heterothallic fungi [60,61,62,63,64,65]. Fungi use various reproductive strategies (e.g., sexual, asexual, and parasexual) each with distinct benefits. Sexual reproduction enhances genetic diversity, promotes adaptation, eliminates harmful mutations, selects beneficial ones, and produces durable, resistant spores capable of long-term survival in unfavorable conditions. In most ascomycetes, sexual spores are dispersed by wind, facilitating gene flow over long distances [64,66]. Given the critical role of reproductive mode in shaping population genetics, evolution, and fungal pathogen management, future research should explore the distribution and function of mating-type genes in *Alternaria* section *Nimbya* isolates.

With molecular biology and technology advancements, new methods have become widely adopted in fungal taxonomy and systematics. Based on molecular biology, the phylogenetic species identification approach addresses some limitations of morphological species identification, offering a more scientific means of understanding fungal phylogenetic relationships. It also provides a reliable foundation and technical methodologies for rapid molecular detection and strain identification [65,67,68]. Recent studies indicate that the *Alternaria* section *Nimbya* comprises numerous species, including cryptic ones that can be distinguished through phylogenetic analysis [20]. In this study, a combination of morphological characteristics, phylogenetic analyses, and Pairwise Homoplasy Index (PHI) tests was used to identify new species within the *Nimbya* section. Phylogenetic analysis based on a five-gene dataset strongly confirmed the distinctiveness of the identified species, supported by robust monophyletic statistical values. In addition, the PHI test supported the results of morphological characteristics and phylogenetic analyses. The findings of this study highlight the high diversity of *Alternaria* species within the *Nimbya* section, associated with the plants in the families of *Cyperaceae* and *Juncaceae* (Table 4). Further studies in other regions could determine the exact diversity of *Alternaria* species in the section *Nimbya* from these plants, the geographical distribution status of the species, and a better understanding of the host relationships.

*Cyperaceae* and *Juncaceae* are two well-established families within the order *Poales*, playing major roles in ecological, economic, and ethnobotanical contexts. These families are particularly dominant in wetland ecosystems [25,69], where they provide essential food and habitat for numerous animal and fungal species with specialized interrelationships. The decline of each plant species within these families could disrupt the associated biota and other dependent organisms [70]. Among the key threats to their growth, reproduction, and survival are fungal pathogens. Understanding the diversity of fungi linked to these plants and their host specificity is essential for effective conservation efforts. Conversely, members of the *Cyperaceae* family have also been recognized as invasive weeds, posing significant challenges to natural ecosystems, agriculture, and forestry [26,71,72]. Weeds compete for vital resources such as water, nutrients, and sunlight, ultimately reducing both the quantity and quality of agricultural products. While chemical herbicides are widely used for weed control, their extensive application has led to issues like environmental pollution, negative impacts on human and animal health, harm to cultivated plants, and the development of herbicide-resistant weed populations. As a result, sustainable and effective weed management alternatives have become a major research focus in recent years. One promising approach is the use of plant-pathogenic fungi as biocontrol agents. Among these, more than ten species of *Alternaria* have been identified as potential candidates for biological weed control [73,74,75,76]. Given the specific host association between species in the *Alternaria* section *Nimbya* and *Cyperaceae* plants, further studies are needed to identify the most suitable and effective fungal species for biocontrol applications.

## Figures and Tables

**Figure 1 jof-11-00225-f001:**
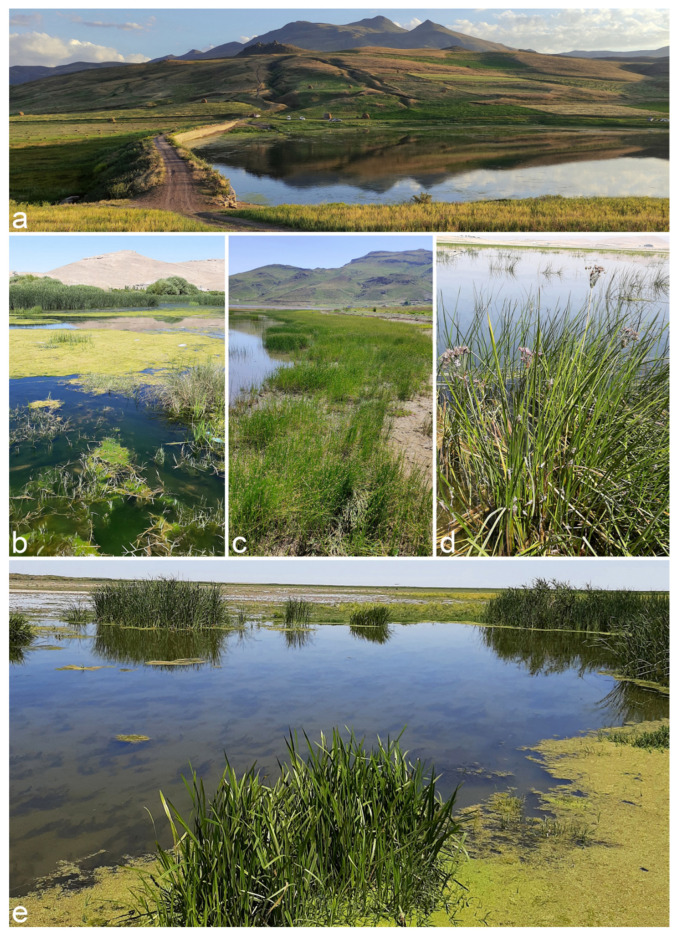
Wetlands sampled in this research. (**a**) Qezeljeh Dam wetland in Khoy County, West Azarbaijan; (**b**) Gori Lake (Qurugöl) wetland in Bostanabad County, East Azarbaijan province; (**c**) Neor Lake in Ardebil County, Ardebil province; (**d**) Norouzlu Dam wetland in Miyandoab County, West Azarbaijan; (**e**) Kani Barazan wetland in Mahabad County, West Azarbaijan province.

**Figure 2 jof-11-00225-f002:**
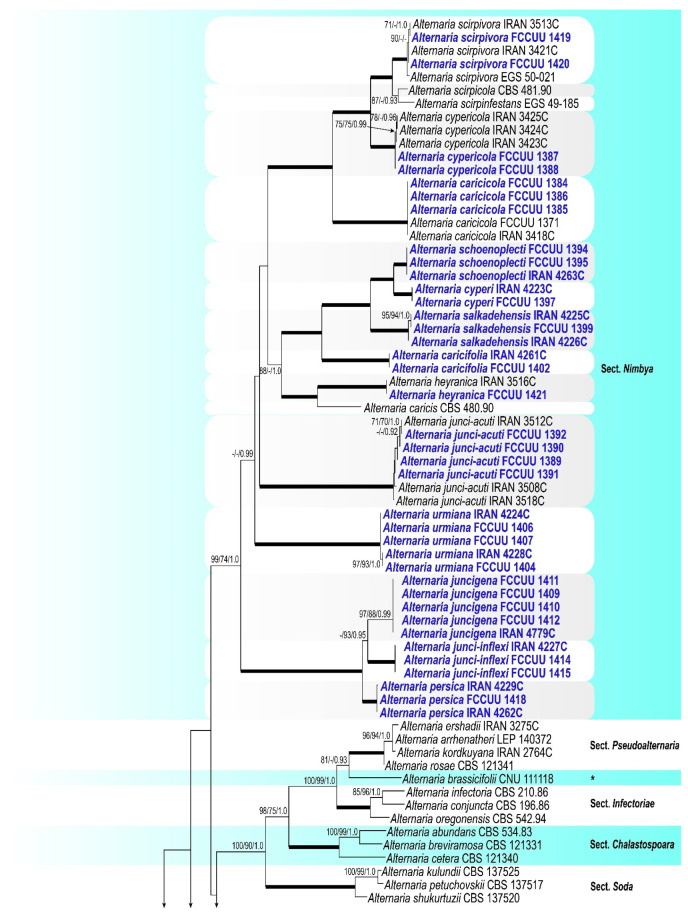
Large-scale phylogenetic tree inferred using Bayesian Inference (BI) based on a combined dataset of ITS, *GAPDH*, *TEF1*, *RPB2*, and *Alt a 1* of *Alternaria* species. Maximum Likelihood (ML), Maximum Parsimony (MP) bootstrap support (BS) values > 70%, and Bayesian posterior probabilities (PP) > 0.90 are indicated at the nodes. The tree is rooted with *Stemphylium botryosum* (CBS 714.68) and *S. vesicarium* (CBS 191.86), and newly identified strains are highlighted in blue. Thickened branches represent the statistical support values of ML/MP/BI analyses equal to 100/100/1.0. The scale bar indicates the number of nucleotide substitutions. The monotypic lineages are indicated by a black asterisk.

**Figure 3 jof-11-00225-f003:**
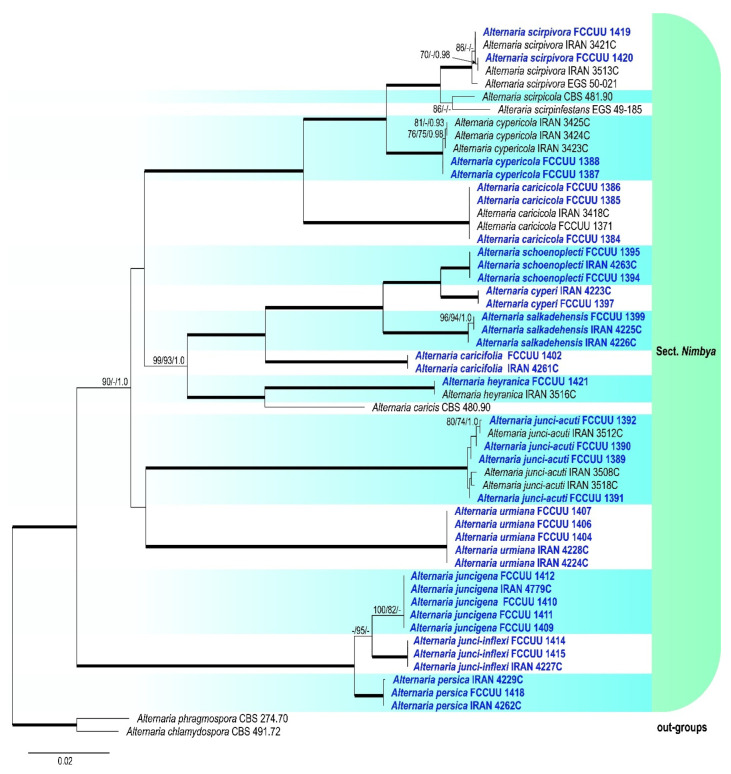
Small-scale phylogenetic tree of *Alternaria* section *Nimbya* species, derived from Bayesian analysis of the combined ITS, *GAPDH*, *TEF1*, *RPB2*, and *Alt a 1* sequence alignment. Maximum likelihood (ML), Maximum Parsimony (MP) bootstrap support values > 70%, and Bayesian posterior probabilities (PP > 0.90) are shown at the nodes. Thickened branches represent the statistical support values of ML/MP/BI equal to 100/100/1.0. Taxonomic novelties are highlighted in bold and blue. The tree is rooted with *Alternaria chlamydospora* (CBS 491.72) and *A. phragmospora* (CBS 274.70).

**Table 1 jof-11-00225-t001:** Primer sets used for PCR amplifications in this study, with sequences and references.

Loci	Primer Name	Primer Sequence (5′–3′)	Direction	Reference
ITS	ITS1	TCCGTAGGTGAACCTGCGG	Forward	[32]
	ITS4	TCCTCCGCTTATTGATATGC	Reverse	
*GAPDH*	gpd1	CAACGGCTTCGGTCGCATTG	Forward	[33]
	gpd2	GCCAAGCAGTTGGTTGTG	Reverse	
*TEF1*	EF1-728F	CATCGAGAAGTTCGAGAAGG	Forward	[34]
	EF1-986R	TACTTGAAGGAACCCTTACC	Reverse	
*RPB2*	RPB2-5F2	GGGGWGAYCAGAAGAAGGC	Forward	[35]
	RPB2-7cR	CCCATRGCTTGTYYRCCCAT	Reverse	[36]
*Alt a 1*	Alt-for	ATGCAGTTCAC CACCATCGC	Forward	[37]
	Alt-rev	ACGAGGGTGAY GTAGGCGTC	Reverse	
ISSR	ISSR5	(GA)_5_YC	-	-

**Table 2 jof-11-00225-t002:** Lists of *Alternaria* species and sections used for phylogenetic analyses, with details about host/substrate, country, and GenBank accession numbers. Newly generated sequences are in bold.

Species Name	Section	Collection No.	Country	Host/Substrate	GenBank Accession Numbers				
					ITS	*GAPDH*	*TEF1*	*RPB2*	*Alt a 1*
** *Alternaria abundans* **	*Chalastospora*	CBS 534.83	New Zealand	*Fragaria* sp.	JN383485	KC584154	KC584707	KC584448	JN383503
** *A. alternantherae* **	*Althernantherae*	CBS 124392	China	*Solanum melongena*	KC584179	KC584096	KC584633	KC584374	KP123846
** *A. alternariae* **	*Ulocladium*	CBS 126989	USA	*Daucus carota*	AF229485	AY278815	KC584730	KC584470	AY563316
** *A. alternata* **	*Alternaria*	CBS 916.96	India	*Arachis hypogaea*	AF347031	AY278808	KC584634	KC584375	AY563301
** *A. arborescens* **	*Alternaria*	CBS 102605	USA	*Solanum lycopersicum*	AF347033	AY278810	KC584636	KC584377	AY563303
** *A. argyranthemi* **	*-*	CBS 116530	New Zealand	*Argyranthemum* sp.	KC584181	KC584098	KC584637	KC584378	-
** *A. arrhenatheri* **	*Pseudoalternaria*	LEP 140372	USA	*Arrhenatherum elatius*	JQ693677	JQ693635	-	-	-
** *A. aspera* **	*Pseudoulocladium*	CBS 115269	Japan	*Pistacia vera*	KC584242	KC584166	KC584734	KC584474	KF533899
** *A. bornmuelleri* **	*Undifilum*	DAOM 231361	Austria	*Securigera varia*	FJ357317	FJ357305	KC584751	KC584491	JN383516
** *A. botryospora* **	*Embellisioides*	CBS_478.90	New Zealand	*Leptinella dioica*	AY278844	AY278831	KC584720	KC584461	AY563324
** *A. botrytis* **	*Ulocladium*	CBS 197.67	USA	Contaminant	KC584243	KC584168	KC584736	KC584476	-
** *A. brassicae* **	*-*	CBS 116528	USA	*Brassica oleracea*	KC584185	KC584102	KC584641	KC584382	-
** *A. brassicicola* **	*Brassicicola*	CBS 118699	USA	*Brassica oleracea*	JX499031	KC584103	KC584642	KC584383	-
** *A. brassicifolii* **	*-*	CNU 111118	Korea	*Brassica rapa* L. subsp. *pekinensis*	JQ317188	KM821537	-	-	-
** *A. breviramosa* **	*Chalastospora*	CBS 121331	Australia	*Triticum* sp.	FJ839608	KC584148	KC584700	KC584442	-
** *A. caricicola* **	*Nimbya*	IRAN 3418C	Iran	*Carex* sp.	MK508871	MK505392	MT187265	MT187279	MT187233
** *A. caricicola* **	*Nimbya*	FCCUU 1371	Iran	*Carex* sp.	MK508872	MK505393	MT187266	MT187280	MT187234
** *A. caricicola* **	** *Nimbya* **	**FCCUU 1384**	**Iran**	***Cyperus* sp.**	**OM349100**	**OM640668**	**OM640735**	**OM640719**	**OM616893**
** *A. caricicola* **	** *Nimbya* **	**FCCUU 1385**	**Iran**	***Cyperus* sp.**	**OM349115**	**OM640669**	**OM640734**	**OM640718**	**OM616894**
** *A. caricicola* **	** *Nimbya* **	**FCCUU 1386**	**Iran**	***Cyperus* sp.**	**OM349116**	**OM640670**	**OM640736**	**OM640720**	**OM616895**
** *A. caricifolia* **	*Nimbya*	IRAN 4261C	Iran	*Cyperus* sp.	OM349113	OM640666	OM640748	OM640714	OM616920
** *A. caricifolia* **	*Nimbya*	FCCUU 1402	Iran	*Cyperus* sp.	OM349114	OM640667	OM640749	OM640715	OM616921
** *A. caricis* **	*Nimbya*	CBS 480.90	USA	*Carex hoodii*	AY278839	AY278826	KC584726	KC584467	AY563321
** *A. cetera* **	*Chalastospora*	CBS 121340	Australia	*Elymus scabrus*	JN383482	AY562398	KC584699	KC584441	AY563278
** *A. chartarum* **	*Pseudoulocladium*	CBS 200.67	Canada	*Populus* sp.	AF229488	KC584172	KC584741	KC584481	AY563319
** *A. cheiranthi* **	*Cheiranthus*	CBS 109384	Italy	*Cheiranthus cheiri*	AF229457	KC584107	KC584646	KC584387	AY563290
** *A. chlamydospora* **	*Phragmosporae*	CBS 491.72	Egypt	Soil	KC584189	KC584108	KC584647	KC584388	-
** *A. chlamydosporigena* **	*Embellisia*	CBS 341.71	USA	Air	KC584231	KC584156	KC584710	KC584451	-
** *A. cinerariae* **	*Sonchi*	CBS 116495	USA	*Ligularia* sp.	KC584190	KC584109	KC584648	KC584389	-
** *A. conjuncta* **	*Infectoriae*	CBS 196.86	Switzerland	*Pastinaca sativa*	FJ266475	AY562401	KC584649	KC584390	AY563281
** *A. conoidea* **	*Brassicicola*	CBS 132.89	Saudi Arabia	*Ricinus communis*	AF348226	FJ348227	KC584711	KC584452	FJ348228
** *A. cucurbitae* **	*Ulocladioides*	CBS 483.81	New Zealand	*Cucumis sativus*	FJ266483	AY562418	KC584743	KC584483	AY563315
** *A. cumini* **	*Eureka*	CBS 121329	India	*Cuminum cyminum*	KC584191	KC584110	KC584650	KC584391	-
** *A. cyperi* **	** *Nimbya* **	**IRAN 4223C**	**Iran**	***Cyperus* sp.**	**OM349106**	**OM640676**	**OM640745**	**OM640712**	**OM616918**
** *A. cyperi* **	** *Nimbya* **	**FCCUU 1397**	**Iran**	***Cyperus* sp.**	**OM349107**	**OM640677**	**OM640746**	**OM640713**	**OM616919**
** *A. cypericola* **	*Nimbya*	IRAN 3423C	Iran	*Cyperus* sp.	MT176120	MT187250	MT187262	MT187276	MT187235
** *A. cypericola* **	*Nimbya*	IRAN 3424C	Iran	*Cyperus* sp.	MT176122	MT187251	MT187263	MT187277	MT187236
** *A. cypericola* **	** *Nimbya* **	**FCCUU 1387**	**Iran**	***Cyperus* sp.**	**OM349101**	**-**	**-**	**OM640716**	**OM616926**
** *A. cypericola* **	** *Nimbya* **	**FCCUU 1388**	**Iran**	***Eleocharis* sp.**	**OM349102**	**-**	**-**	**OM640717**	**OM616927**
** *A. cypericola* **	*Nimbya*	IRAN 3425C	Iran	*Cyperus* sp.	MT176121	MT187252	MT187264	MT187278	MT187237
** *A. daucifolii* **	*Alternata*	CBS 118812	USA	*Daucus carota*	KC584193	KC584112	KC584652	KC584393	KP123905
** *A. dennisii* **	*-*	CBS 476.90	Isle of Man	*Senecio jacobaea*	JN383488	JN383469	KC584713	KC584454	JN383505
** *A. dianthicola* **	*Dianthicola*	CBS 116491	New Zealand	*Dianthus* × *allwoodii*	KC584194	KC584113	KC584653	KC584394	-
** *A. elegans* **	*Dianthicola*	CBS 109159	Burkina Faso	*Lycopersicon esculentum*	KC584195	KC584114	KC584654	KC584395	-
** *A. embellisia* **	*Embellisia*	CBS 339.71	USA	*Allium sativum*	KC584230	KC584155	KC584708	KC584449	-
** *A. ershadii* **	*Pseudoalternaria*	IRAN 3275C	Iran	*Triticum aestivum*	MK829647	MK829645	-	-	-
** *A. eryngii* **	*Panax*	CBS 121339	-	*Eryngium* sp.	JQ693661	AY562416	KC584656	KC584397	AY563313
** *A. euphorbiicola* **	*Euphorbiicola*	CBS 133874	USA	*Euphorbia hyssopifolia*	KJ718174	KJ718019	KJ718522	KJ718347	-
** *A. euphorbiicola* **	*Euphorbiicola*	CBS 119410	USA	*Euphorbia pulcherrima*	KJ718173	KJ718018	KJ718521	KJ718346	-
** *A. eureka* **	*Eureka*	CBS 193.86	Australia	*Medicago rugosa*	JN383490	JN383471	KC584715	KC584456	JN383507
** *A. gypsophilae* **	*Gypsophilae*	CBS 107.41	Netherlands	*Gypsophila elegans*	KC584199	KC584118	KC584660	KC584401	KJ718688
** *A. helianthiinficiens* **	*helianthiinficiens*	CBS 208.86	USA	*Helianthus annuus*	JX101649	KC584120	EU130548	KC584403	-
** *A. heterospora* **	*Ulocladioides*	CBS 123376	China	*Lycopersicon esculentum*	KC584248	KC584176	KC584748	KC584488	EU855805
** *A. heyranica* **	*Nimbya*	IRAN 3516C	Iran	*Carex* sp.	MT176114	MT187244	MT187256	MT187270	MT187238
** *A. heyranica* **	** *Nimbya* **	**FCCUU 1421**	**Iran**	***Carex* sp.**	**OM535905**	**OM640671**	**OM640758**	**OM640726**	**OM616930**
** *A. hyacinthi* **	*Embellisioides*	CBS 416.71	Netherlands	*Hyacinthus orientalis*	KC584233	KC584158	KC584716	KC584457	-
** *A. indefessa* **	*Cheiranthus*	CBS 536.83	USA	Soil	KC584234	KC584159	KC584717	KC584458	AY563323
** *A. infectoria* **	*Infectoriae*	CBS 210.86	UK	*Triticum aestivum*	DQ323697	AY278793	KC584662	KC584404	FJ266502
** *A. japonica* **	*Japonicae*	CBS 118390	USA	*Brassica chinensis*	KC584201	KC584121	KC584663	KC584405	-
** *A. junci-acuti* **	*Nimbya*	IRAN 3508C	Iran	*Juncus acutus*	MT176111	MT187241	MT187253	MT187267	MT187230
** *A. junci-acuti* **	*Nimbya*	IRAN 3518C	Iran	*Juncus acutus*	MT176112	MT187242	MT187254	MT187268	MT187232
** *A. junci-acuti* **	*Nimbya*	IRAN 3512C	Iran	*Juncus acutus*	MT176113	MT187243	MT187255	MT187269	MT187231
** *A. junci-acuti* **	** *Nimbya* **	**FCCUU 1389**	**Iran**	***Carex* sp.**	**OM349091**	**OM640674**	**-**	**OM640704**	**OM616923**
** *A. junci-acuti* **	** *Nimbya* **	**FCCUU 1390**	**Iran**	***Carex* sp.**	**OM349092**	**OM640672**	**-**	**OM640702**	**OM616922**
** *A. junci-acuti* **	** *Nimbya* **	**FCCUU 1391**	**Iran**	***Carex* sp.**	**OM349093**	**OM640675**	**-**	**OM640705**	**OM616924**
** *A. junci-acuti* **	** *Nimbya* **	**FCCUU 1392**	**Iran**	***Carex* sp.**	**OM349090**	**OM640673**	**-**	**OM640703**	**OM616925**
** *A. juncigena* **	** *Nimbya* **	**IRAN 4779C**	**Iran**	***Juncus* sp.**	**OM349085**	**OM640661**	**OM640740**	**OM640697**	**OM616904**
** *A. juncigena* **	** *Nimbya* **	**FCCUU 1409**	**Iran**	***Juncus* sp.**	**OM349086**	**OM640662**	**OM640741**	**OM640698**	**OM616905**
** *A. juncigena* **	** *Nimbya* **	**FCCUU 1410**	**Iran**	***Juncus* sp.**	**OM349087**	**OM640663**	**OM640742**	**OM640699**	**OM616906**
** *A. juncigena* **	** *Nimbya* **	**FCCUU 1411**	**Iran**	***Juncus* sp.**	**OM349088**	**OM640664**	**OM640743**	**OM640700**	**OM616907**
** *A. juncigena* **	** *Nimbya* **	**FCCUU 1412**	**Iran**	***Juncus* sp.**	**OM349089**	**OM640665**	**OM640744**	**OM640701**	**OM616908**
** *A. junci-inflexi* **	** *Nimbya* **	**IRAN 4227C**	**Iran**	** *Juncus inflexus* **	**OM349097**	**OM640660**	**OM640752**	**OM640696**	**OM616912**
** *A. junci-inflexi* **	** *Nimbya* **	**FCCUU 1414**	**Iran**	** *Juncus inflexus* **	**OM349098**	**OM640659**	**OM640751**	**OM640695**	**OM616914**
** *A. junci-inflexi* **	** *Nimbya* **	**FCCUU 1415**	**Iran**	** *Juncus inflexus* **	**OM349099**	**OM640658**	**OM640750**	**OM640694**	**OM616913**
** *A. kordkuyana* **	*Pseudoalternaria*	IRAN 2764C	Iran	*Triticum aestivum*	MF033843	MF033826	-	-	-
** *A. kulundii* **	*Soda*	CBS 137525	Russia	Soil	KJ443262	KJ649618	-	KJ443176	-
** *A. leucanthemi* **	*Teretispora*	CBS 421.65	Netherlands	*Chrysanthemum maximum*	KC584240	KC584164	KC584732	KC584472	-
** *A. leucanthemi* **	*Teretispora*	CBS 422.65	USA	*Chrysanthemum maximum*	KC584241	KC584165	KC584733	KC584473	-
** *A. mimicula* **	*Brassicicola*	CBS 118696	USA	*Lycopersicon esculentum*	FJ266477	AY562415	KC584669	KC584411	AY563310
** *A. nepalensis* **	*Japonicae*	CBS 118700	Nepal	*Brassica* sp.	KC584207	KC584126	KC584672	KC584414	-
** *A. nobilis* **	*Gypsophilae*	CBS 116490	New Zealand	*Dianthus caryophyllus*	KC584208	KC584127	KC584673	KC584415	JQ646385
** *A. omanensis* **	*Omanensis*	SQUCC 15560	Oman	dead wood	MK878563	MK880900	MK880897	MK880894	-
** *A. omanensis* **	*Omanensis*	SQUCC 13580	Oman	dead wood	MK878562	MK880899	MK880896	MK880893	-
** *A. oregonensis* **	*Infectoriae*	CBS 542.94	USA	*Triticum aestivum*	FJ266478	FJ266491	KC584674	KC584416	FJ266503
** *A. oudemansii* **	*Ulocladium*	CBS 114.07	-	-	FJ266488	KC584175	KC584746	KC584486	FJ266514
** *A. panax* **	*Panax*	CBS 482.81	USA	*Aralia racemosa*	KC584209	KC584128	KC584675	KC584417	JQ646382
** *A. penicillata* **	*Crivellia*	CBS 116607	Austria	*Papaver rhoeas*	KC584229	KC584153	KC584706	KC584447	-
** *A. penicillata* **	*Crivellia*	CBS 116608	Austria	*Papaver rhoeas*	FJ357311	FJ357299	KC584698	KC584440	JN383502
** *A. perpunctulata* **	*Althernantherae*	CBS 115267	USA	*Alternanthera philoxeroides*	KC584210	KC584129	KC584676	KC584418	JQ905111
** *A. persica* **	** *Nimbya* **	**IRAN 4229C**	**Iran**	***Juncus* sp.**	**OM349082**	**OM640655**	**OM640737**	**OM640691**	**OM616901**
** *A. persica* **	** *Nimbya* **	**IRAN 4262C**	**Iran**	***Juncus* sp.**	**OM349083**	**OM640656**	**OM640738**	**OM640692**	**OM616902**
** *A. persica* **	** *Nimbya* **	**FCCUU 1418**	**Iran**	***Juncus* sp.**	**OM349084**	**OM640657**	**OM640739**	**OM640693**	**OM616903**
** *A. petroselini* **	*Radicina*	CBS 112.41	-	*Petroselinum sativum*	KC584211	KC584130	KC584677	KC584419	AY563288
** *A. petuchovskii* **	*Soda*	CBS 137517	Russia	Soil	KJ443254	KJ649616	-	KJ443170	-
** *A. photistica* **	*Panax*	CBS 212.86	UK	*Digitalis purpurea*	KC584212	KC584131	KC584678	KC584420	AY563282
** *A. phragmospora* **	*Phragmosporae*	CBS 274.70	Netherlands	Soil	JN383493	JN383474	KC584721	KC584462	JN383509
** *A. porri* **	*Porri*	CBS 116699	USA	*Allium cepa*	KJ718218	KJ718053	KJ718564	KJ718391	KJ718727
** *A. proteae* **	*Embellisioides*	CBS 475.90	Australia	*Protea* sp.	AY278842	KC584161	KC584723	KC584464	-
** *A. protenta* **	*Porri*	CBS 116651	USA	*Solanum tuberosum*	KC584217	KC584139	KC584688	KC584430	GQ180097
** *A. pseudorostrata* **	*Porri*	CBS 119411	USA	*Euphorbia pulcherrima*	JN383483	AY562406	KC584680	KC584422	AY563295
** *A. radicina* **	*Radicina*	CBS 245.67	USA	*Daucus carota*	KC584213	KC584133	KC584681	KC584423	FN689405
** *A. rosae* **	*Pseudoalternaria*	CBS 121341	New Zealand	*Rosa rubiginosa*	JQ693639	JQ646279	-	-	JQ646370
** *A. salkadehensis* **	** *Nimbya* **	**IRAN 4226C**	**Iran**	***Carex* sp.**	**OM349094**	**OM640690**	**OM640747**	**OM640706**	**OM616911**
** *A. salkadehensis* **	** *Nimbya* **	**FCCUU 1399**	**Iran**	***Cyperus* sp.**	**OM349095**	**OM640688**	**OM640732**	**OM640707**	**OM616909**
** *A. salkadehensis* **	** *Nimbya* **	**IRAN 4225C**	**Iran**	***Cyperus* sp.**	**OM349096**	**OM640689**	**OM640733**	**OM640708**	**OM616910**
** *A. schoenoplecti* **	** *Nimbya* **	**IRAN 4263C**	**Iran**	***Schoenoplectu* sp.**	**OM349103**	**OM640678**	**OM640729**	**OM640709**	**OM616915**
** *A. schoenoplecti* **	** *Nimbya* **	**FCCUU 1394**	**Iran**	***Schoenoplectu* sp.**	**OM349104**	**OM640679**	**OM640730**	**OM640710**	**OM616916**
** *A. schoenoplecti* **	** *Nimbya* **	**FCCUU 1395**	**Iran**	***Schoenoplectu* sp.**	**OM349105**	**OM640680**	**OM640731**	**OM640711**	**OM616917**
** *A. scirpicola* **	*Nimbya*	CBS 481.90	UK	*Scirpus* sp.	KC584237	KC584163	KC584728	KC584469	-
** *A. scirpinfestans* **	*Nimbya*	EGS 49-185	USA	*Scirpus acutus*	JN383499	JN383480	-	-	JN383514
** *A. scirpivora* **	*Nimbya*	EGS 50-021	USA	*Scirpus acutus*	JN383500	JN383481	-	-	JN383515
** *A. scirpivora* **	** *Nimbya* **	**FCCUU 1419**	**Iran**	** *Scirpus acutus* **	**OM535903**	**OM640681**	**OM640759**	**OM640727**	**OM616928**
** *A. scirpivora* **	** *Nimbya* **	**FCCUU 1420**	**Iran**	** *Scirpus acutus* **	**OM535904**	**OM640682**	**OM640760**	**OM640728**	**OM616929**
** *A. scirpivora* **	*Nimbya*	IRAN 3421C	Iran	*Scirpus acutus*	MT176118	MT187248	MT187260	MT187274	MT187240
** *A. scirpivora* **	*Nimbya*	IRAN 3419C	Iran	*Scirpus acutus*	MT176119	MT187249	MT187261	MT187275	-
** *A. septospora* **	*Pseudoulocladium*	CBS 109.38	Italy	Wood	FJ266489	FJ266500	KC584747	KC584487	FJ266515
** *A. shukurtuzii* **	*Soda*	CBS 137520	Russia	Soil	KJ443257	KJ649620	-	KJ443172	-
** *A. simsimi* **	*Dianthicola*	CBS 115265	Argentina	*Sesamum indicum*	JF780937	KC584137	KC584686	KC584428	-
** *A. smyrnii* **	*Radicina*	CBS 109380	UK	*Smyrnium olusatrum*	AF229456	KC584138	KC584687	KC584429	AY563289
** *A. soliaridae* **	*-*	CBS 118387	USA	Soil	KC584218	KC584140	KC584689	KC584431	-
** *A. sonchi* **	*Sonchi*	CBS 119675	Canada	*Sonchus asper*	KC584220	KC584142	KC584691	KC584433	-
** *A. tellustris* **	*Embellisia*	CBS 538.83	USA	Soil	FJ357316	AY562419	KC584724	KC584465	AY563325
** *A. thalictrigena* **	*-*	CBS 121712	Germany	*Thalictrum* sp.	EU040211	KC584144	KC584694	KC584436	-
** *A. triglochinicola* **	*Eureka*	CBS 119676	Australia	*Triglochin procera*	KC584222	KC584145	KC584695	KC584437	-
** *A. urmiana* **	** *Nimbya* **	**IRAN 4228C**	**Iran**	** *Juncus inflexus* **	**OM349108**	**OM640683**	**OM640754**	**OM640721**	**OM616896**
** *A. urmiana* **	** *Nimbya* **	**FCCUU 1404**	**Iran**	** *Juncus inflexus* **	**OM349109**	**OM640684**	**OM640755**	**OM640722**	**OM616897**
** *A. urmiana* **	** *Nimbya* **	**IRAN 4224C**	**Iran**	** *Juncus acutus* **	**OM349110**	**OM640685**	**OM640753**	**OM640723**	**OM616898**
** *A. urmiana* **	** *Nimbya* **	**FCCUU 1406**	**Iran**	** *Juncus acutus* **	**OM349111**	**OM640686**	**OM640756**	**OM640724**	**OM616899**
** *A. urmiana* **	** *Nimbya* **	**FCCUU 1407**	**Iran**	** *Juncus acutus* **	**OM349112**	**OM640687**	**OM640757**	**OM640725**	**OM616900**
** *A. vaccariicola* **	*Gypsophilae*	CBS 118714	USA	*Vaccaria hispanica*	KC584224	KC584147	KC584697	KC584439	JQ646384
** *Stemphylium botryosum* **	*-*	CBS 714.68	Canada	*Medicago sativa*	KC584238	AF443881	KC584729	AF107804	-
** *S. vesicarium* **	*-*	CBS 191.86	India	*Medicago sativa*	KC584239	AF443884	KC584731	KC584471	-

**Table 3 jof-11-00225-t003:** Phylogenetic information of individual and combined sequence datasets used in phylogenetic analyses.

Gene		Parameter								
		Number of Taxa	Total Characters	Constant Sites	Variable Sites	Parsimony Informative Sites	Parsimony Uninformative Sites	AIC Substitution Model *	Lset Nst, Rates	−lnL
Section *Nimbya*	ITS	55	469	388	81	76	5	SYM + I + G	6, invgamma	1393.2854
	*GAPDH*	53	514	368	146	138	8	GTR + I	6, propinv	2019.9746
	*TEF1*	46	177	113	64	60	4	K80 + G	2, gamma	758.6035
	*RPB2*	53	716	526	190	178	12	GTR + G	6, gamma	2521.9597
	*Alt a 1*	50	431	215	216	193	23	GTR + I + G	6, invgamma	2506.1685
	Combined	55	2307	1610	697	645	52	–	–	10,258.9258
All *Alternaria* sections	ITS	132	462	333	129	111	18	GTR + I + G	6, invgamma	2978.5835
	*GAPDH*	130	524	311	213	192	21	GTR + I + G	6, invgamma	5840.7612
	*TEF1*	107	187	59	127	109	18	GTR + G	6, gamma	2732.1633
	*RPB2*	125	690	437	253	247	6	SYM + I + G	6, invgamma	7530.5337
	*Alt a 1*	87	436	117	320	277	43	HKY + I + G	6, invgamma	7424.5391
	Combined	132	2299	1257	1042	936	106	–	–	28,937.1276

* Akaike information criterion substitution models implemented in Bayesian inference.

**Table 4 jof-11-00225-t004:** A comparative analysis of the 13 species identified in this study based on host association, morphological features, and cultural characteristics on PCA medium at 25 °C after 7 days of growth. Measurements for conidial characteristics are given in ranges (minimum–maximum).

Species	Host Family (Genus or Species)	Conidial Characteristics	Colony Characteristics (on PCA)	Sexual Morph	Distribution in Iran (province)
*A. caricicola*	*Cyperaceae* (*Cyperus* sp. and *Carex* sp.)	87–205 × 10–15 μm; 7–14 transverse septa; smooth; Beaks 30–100	Flat, entire, olivaceous green to grey olivaceous; 60–70 mm/7d	Not observed	East Azarbaijan; West Azarbaijan
*A. caricifolia*	*Cyperaceae* (*Carex* sp.)	35–100 × 5–9 μm; 3–11 transverse septa; smooth, apical cell extension up to 45 μm	Flat, floccose, white to rosy buff center, hazel margins; 58 mm/7d	Present; ascomata 120–240 × 90–220 μm	West Azarbaijan
*A. cyperi*	*Cyperaceae* (*Cyperus* sp.)	55–100 × 10–12 μm; 5–12 transverse septa; smooth; apical cell extension up to 15 μm	Fawn with salmon tints; velvety; 59 mm/7d	Not observed	East Azarbaijan
*A. cypericola*	*Cyperaceae* (*Cyperus* sp. and *Eleocharis* sp.)	87–155 × 10–15 μm; 6–12 transverse septa; smooth; Beaks 37.5–137	Flat, entire, dark green to olivaceous brown; 60–70 mm/7d	Not observed	West Azarbaijan
*A. heyranica*	*Cyperaceae* (*Carex* sp.)	50–75 × 10–12 μm; 4–9 transverse septa; smooth, Beaks 50–175	Flat, entire, olivaceous brown; 25–30 mm/7d	Not observed	Guilan
*A. junci–acuti*	*Cyperaceae* (*Carex* sp.); *Juncaceae* (*Juncus acutus*)	50–87 × 7–9 μm; 5–14 transverse septa; smooth, Beaks 20–200	Flat, entire, dark green to olivaceous brown; 70–80 mm/7d	Not observed	Golestan; West Azarbaijan
*A. juncigena* sp. nov.	*Juncaceae* (*Juncus* sp.)	40–110 × 10–16 μm; 5–15 transverse septa; smooth; apical cell extension up to 50 μm	Flat, entire, velvety, pale vinaceous to vinaceous buff; 50 mm/7d	Not observed	West Azarbaijan
*A. junci–inflexi*	*Juncaceae* (*Juncus inflexus*)	21–80 × 8–12 μm; 2–9 transverse septa; smooth to verrucose; apical cell extension up to 15 μm	Flat, entire, velvety, fawn; 65 mm/7d	Not observed	East Azarbaijan; West Azarbaijan
*A. persica*	*Juncaceae* (*Juncus* sp.)	40–85 × 8–11 μm; 4–15 transverse septa; smooth to verrucose; apical cell extension up to 50 μm	Floccose, vinaceous buff; 60 mm/7d	Not observed	West Azarbaijan
*A. salkadehensis*	*Cyperaceae* (*Cyperus* sp. and *Carex* sp.)	35–100 × 10–13 μm; 4–13 transverse septa, smooth; apical cell extension up to 45 μm	Flat, entire, velvety, hazel; 63 mm/7d	Not observed	West Azarbaijan
*A. schoenoplecti*	*Cyperaceae* (*Schoenoplectus* sp.)	40–88 × 12–18 μm; 3–11 transverse septa; smooth; apical cell extension up to 22 μm	Flat, entire, velvety, sepia to cinnamon; 49 mm/7d	Not observed	West Azarbaijan
*A. scirpivora*	*Cyperaceae* (*Scirpus acutus*)	30–60 × 5–8 μm; 3–10 transverse septa; smooth; apical cell extension up to 50 μm	Flat, entire, dark green to olivaceous brown; 60–70 mm/7d	Present; ascomata 280–500 × 250–450 μm	Multiple provinces
*A. urmiana*	*Juncaceae* (*Juncus acutus* and *Juncus inflexus*)	35–55 × 12–15 μm; 3–8 transverse septa; smooth to verrucose; apical cell extension up to 10 μm	Flat, entire, velvety, rosy buff; 51 mm/7d	Not observed	West Azarbaijan

## Data Availability

The original contributions presented in this study are included in the article and Appendix A. Further inquiries can be directed to the corresponding authors.

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
