# Peer review of "Diversity of Alternaria Section Nimbya in Iran, with the Description of Eight New Species"

_jof, 2025, doi:10.3390/jof11030225_

Round 1

Reviewer 1 Report

This study is useful to fungal taxonomy. The methods used in this study are sound and solid.  For the minor issues, please refer to the annotated manuscript.

It is my pleasure to review the manuscript "Diversity of Alternaria Section Nimbya in Iran, with the Description of Eight New Species" (manuscript no. jof-3527691). The manuscript is well written. There are some minor issues and no major issues. Microphotograph plates are well prepared with nice photos. The manuscript has been annotated. For the minor issues, please refer to the annotated manuscript.
Other issues are 1) replace “additional isolates examines” of new species with “additional specimen examined”, and add the number of paratype corresponding to the culture. 2) In Materials examined, Please add the number of specimen corresponding to the culture. 3) For the known species, I think that you should compare their nucleotide sequences with the type strains. 

Author Response

The authors gratefully acknowledge the respected reviewer for the accuracy, suggestions, and comments, which improved the manuscript’s quality. We read the manuscript, accepted, and did all the corrections as suggested, as well as some minor corrections we found, all of them are marked with track changes. Also, the point-by-point responses/answers to the questions are provided below.

Reviewer 1:

Q/R1- replace “additional isolates examined” of new species with “additional specimen examined”.

Answer: It was replaced in the manuscript.

Q/R2- add the number of paratype corresponding to the culture.

Answer: The numbers of 38 isolates which we used in multi-gene phylogenetic analyses, are presented in the text, as IRAN or FCCUU numbers.

Q/R3- In Materials examined, please add the number of specimens corresponding to the culture.

Answer: The numbers of 38 isolates which we used in multi-gene phylogenetic analyses, are presented in the text, as IRAN or FCCUU numbers.

Q/R4- For the known species, I think that you should compare their nucleotide sequences with the type strains.

Answer: nucleotide comparisons with the type strains for the known species were done and highlighted in yellow color.

Reviewer 2 Report

This paper is a comprehensive review of Alternaria section Nimbya with the description of new species isolate from two groups of plants from multiple areas of Iran. Nimbya was long overdue for a rigorous review and comparison. This is excellent work that includes very nice photographs of the host plants, fungi on multiple media and many spore photos showing a range of sizes and shapes for each new species described. Impressive work.

Trivial points to help the manuscript.

Please explain (define) the last few rows of Table 3 in the legend under the table. It is almost impossible to understand.

Please show the data for the ISSR-PCR data. Perhaps in a supplemental table of +/- characters.

Figures 6 and  9 - legend -please clarify since all entries appear to be in bold blue?

Author Response

The authors gratefully acknowledge the respected reviewer for the accuracy, suggestions, and comments, which improved the manuscript’s quality. We read the manuscript, accepted, and did all the corrections as suggested, as well as some minor corrections we found, all of them are marked with track changes. Also, the point-by-point responses/answers to the questions are provided below.

Reviewer 2:

Q/R1- Please explain (define) the last few rows of Table 3 in the legend under the table. It is almost impossible to understand.

Answer: Yes, you are right. As the table is compressed at the horizontal zone, the words placed at different horizons, making them hard to understand. We corrected it.

Q/R2- Please show the data for the ISSR-PCR data. Perhaps in a supplemental table of +/- characters.

Answer: The figures of ISSR-PCR data were added as a supplement file (Supplementary Figure 1).

Q/R3- Figures 6 and 9 - legend -please clarify since all entries appear to be in bold blue?

Answer: As all the species presented in Figures 6 and 9 are new, they are seen in blue bold. For this, we added another species and corrected accordingly.

Q/R4- Please harmonize the order with these five gene concatenated dataset.

Answer: All of them are harmonized based on the recommendation and corrections were done accordingly.

Q/R5- check the reference. Crous et al. 2004 or "Mycobank. Available online: http://www.mycobank.org (accessed on ???date 2025)."

Answer: The reference is correct as it was presented.